# A transient cortical state with sleep-like sensory responses precedes emergence from general anesthesia in humans

**Laura D Lewis[1,2]\*, Giovanni Piantoni[3], Robert A Peterfreund[4,5], Emad N Eskandar[6], Priscilla Grace Harrell[4,5], Oluwaseun Akeju[4,5], Linda S Aglio[5,7], Sydney S Cash[3], Emery N Brown[4,5,8,9], Eran A Mukamel[10]\*, Patrick L Purdon[4,5]\***

[1]Society of Fellows, Harvard University, Cambridge, United States; [2]Athinoula A. Martinos Center for Biomedical Imaging, Department of Radiology, Massachusetts General Hospital, Boston, United States; [3]Department of Neurology, Massachusetts General Hospital and Harvard Medical School, Boston, United States; [4]Anesthesiology, Critical Care, and Pain Medicine, Massachusetts General Hospital, Boston, United States; [5]Harvard Medical School, Boston, United States; [6]Department of Neurosurgery, Massachusetts General Hospital and Harvard Medical School, Boston, United States; [7]Department of Anesthesiology, Perioperative and Pain Medicine, Brigham and Women's Hospital, Boston, United States; [8]Brain and Cognitive Sciences, Massachusetts Institute of Technology, Cambridge, Unites States; [9]Institute for Medical Engineering and Sciences, Massachusetts Institute of Technology, Cambridge, United States; [10]Department of Cognitive Science, University of California, San Diego, San Diego, United States

**Abstract** During awake consciousness, the brain intrinsically maintains a dynamical state in which it can coordinate complex responses to sensory input. How the brain reaches this state spontaneously is not known. General anesthesia provides a unique opportunity to examine how the human brain recovers its functional capabilities after profound unconsciousness. We used intracranial electrocorticography and scalp EEG in humans to track neural dynamics during emergence from propofol general anesthesia. We identify a distinct transient brain state that occurs immediately prior to recovery of behavioral responsiveness. This state is characterized by large, spatially distributed, slow sensory-evoked potentials that resemble the K-complexes that are hallmarks of stage two sleep. However, the ongoing spontaneous dynamics in this transitional state differ from sleep. These results identify an asymmetry in the neurophysiology of induction and emergence, as the emerging brain can enter a state with a sleep-like sensory blockade before regaining responsivity to arousing stimuli.
DOI: https://doi.org/10.7554/eLife.33250.001

**\*For correspondence:**
ldlewis@nmr.mgh.harvard.edu
(LDL);
emukamel@ucsd.edu (EAM);
patrickp@nmr.mgh.harvard.edu
(PLP)

**Competing interest:** See
page 18

**Reviewing editor:** Saskia
Haegens, Columbia University
College of Physicians and
Surgeons, United States

## Introduction

During emergence from general anesthesia, the brain transitions out of the unconscious state and recovers its ability to process complex sensory information and coordinate behavioral responses. General anesthesia causes a profound disruption of information processing across large-scale cortical (*Alkire et al., 2008*; *Liu et al., 2012*; *Hudetz, 2012*; *Lewis et al., 2012*; *Dehaene et al., 2014*; *Sarasso et al., 2014*) and thalamocortical networks (*Alkire et al., 2000*; *Ching et al., 2010*; *Mhuircheartaigh et al., 2010*; *Ní Mhuircheartaigh et al., 2013*; *Vijayan et al., 2013*; *Akeju et al., 2014*; *Baker et al., 2014*; *Flores et al., 2017*), and suppression of arousal systems (*Lydic and*

**eLife digest** General anesthesia is essential to modern medicine. It allows physicians to temporarily keep people in an unconscious state. When infusions of the anesthetic drug stop, patients gradually recover consciousness and awaken, a process called emergence. Previous studies using recordings of electrical activity in the brain have documented spontaneous changes during anesthesia. In addition, the way the brain responds to sounds or other stimulation is altered. How the brain switches between the anesthetized and awake states is not well understood.

Studying the changes that happen during emergence may help scientists learn how the brain awakens after anesthesia. A key question is whether the changes that occur during emergence are the reverse of what happens when someone is anesthetized, or whether it is a completely different process. Knowing this could help clinicians monitoring patients under anesthesia, and help scientists understand more about how the brain transitions into the awake state.

Now, Lewis et al. show that people go through a sleep-like state right before awakening from anesthesia-induced unconsciousness. In the experiments, recordings were made of the electrical activity in the brains of people emerging from anesthesia. One set of recordings was taken in people with epilepsy, who had electrodes implanted in their brains as part of their treatment. Similar recordings of brain electrical activity during emergence were also made on healthy volunteers using electrodes placed on their scalps. In both groups of people, Lewis et al. documented large changes in electrical activity in the brain's response to sound in the minutes before emergence.

These patterns of electrical activity during emergence were similar to those seen in patients during a normal stage of sleep (stage 2). Patients who were about to wake up from general anesthesia had suppressed brain activity in response to sounds, such as their name. Moreover, this sleep-like state happened only during emergence, indicating it is a distinct process from going under anesthesia. The experiments also suggest that the brain may use a common process to wake up after sleep or anesthesia. More studies may help scientists understand this process and how to better care for patients who need anesthesia.

DOI: https://doi.org/10.7554/eLife.33250.002

*Baghdoyan, 2005*; *Franks, 2008*; *Brown et al., 2011*), which then spontaneously reverts as the anesthetic drug clears. While several studies have characterized the transition in electrophysiological dynamics occurring at loss of consciousness during anesthetic inductions (*Gugino et al., 2001*; *Feshchenko et al., 2004*; *Ferrarelli et al., 2010*; *Supp et al., 2011*; *Lewis et al., 2012*), less is known about the dynamics that occur during emergence from general anesthesia and how they support recovery of a behaviorally responsive state (*Purdon et al., 2013*; *Hudson et al., 2014*; *Mukamel et al., 2014*).

Electrophysiological evidence shows that many anesthetic-induced neurophysiological dynamics undergo relatively symmetric transitions: shifts in spectral power, spatial correlations, phase-amplitude coupling, and spike coherence that are observed during anesthetic induction gradually reverse as drug concentrations are lowered (*Breshears et al., 2010*; *Lee et al., 2010*; *Purdon et al., 2013*; *Mukamel et al., 2014*; *Vizuete et al., 2014*). However, it is also clear that the process of emerging from anesthesia is not identical to anesthetic induction. Emergence occurs at lower anesthetic doses than induction (*Friedman et al., 2010*), and this hysteresis suggests that state-dependent processes also shape the transitions in and out of anesthesia. Behaviorally, some patients experience delirium, a transient state of agitation and confusion which can arise during emergence from anesthesia (*O'Brien, 2002*), suggesting that distinct neural mechanisms may underlie emergence. EEG and local field potential recordings have suggested that the process of emergence may involve stepping through discrete dynamical states (*Lee et al., 2011*; *Hudson et al., 2014*). Electrophysiological studies in rodents show that propofol-induced coherent alpha and delta oscillations, which appear to mediate the functional disruption of thalamus and cortex during anesthesia, recover in a spatiotemporal sequence during emergence that is different from induction (*Flores et al., 2017*). These observations are consistent with a history-dependent process, in which the current brain state influences the process by which the next brain state is reached. However, a neurophysiological mechanism or network dynamic that is engaged selectively during emergence, rather than induction, is not known.

In addition to shifts in spontaneous neurophysiological dynamics, sensory processing is also strongly affected by induction and emergence from general anesthesia. Sensory-evoked potentials (event-related potentials, ERPs) index specific phases of cognitive information processing and can provide diagnostic measures of unconscious patients (*Boly et al., 2011*; *King et al., 2013*). Several studies of ERPs during anesthesia have shown that disruption of higher level cognitive processing is reflected by a reduction in amplitude of the mismatch negativity (MMN), potentials evoked by unexpected sensory input. The MMN declines in amplitude during induction of anesthesia (*Simpson et al., 2002*; *Heinke et al., 2004*), whereas lower-level responses such as the auditory steady-state response persist during sedation, and are abolished at deep anesthetic levels (*Plourde and Picton, 1990*). Cortical responses to direct stimulation using TMS are more spatially constrained and less complex during propofol-induced unconsciousness (*Sarasso et al., 2015*), consistent with fragmentation of large-scale brain network activity during propofol anesthesia (*Lewis et al., 2012*). The propagation of sensory information through thalamocortical circuits is thus differentially affected at increasing doses of anesthesia, with higher-level, longer-latency responses extinguished at low drug levels and then further suppression of short-latency evoked activity at high drug levels.

At the deepest levels of anesthesia, when brain activity enters a state of 'burst suppression' alternating between periods of isoelectric silence (suppressions) and periods of high-amplitude activity (bursts), sensory stimuli can trigger the onset of a burst (*Hartikainen et al., 1995*; *Kroeger and Amzica, 2007*). It is therefore clear that external sensory input can still influence cortical activity during profound anesthesia. However, evoked responses during burst suppression are qualitatively different than those observed during normal sensory processing, as they typically manifest as a large-amplitude burst containing the spectral dynamics of the pre-bursting state (*Lewis et al., 2013*), rather than the distinct ERP waveform with classical components related to specific phases of cognitive information processing seen in the waking state. Sensory input during burst suppression thus appears to drive nonspecific cortical activity rather than effective processing of sensory information.

The neural dynamics supporting the brain's ability to spontaneously recover wakeful consciousness, regain sensory perception and resume complex cortical responses following the profound disruption caused by general anesthesia are not well understood. Late components of the ERP continue to be disrupted even after patients have recovered consciousness and early components have returned to baseline (*Plourde and Picton, 1991*; *Koelsch et al., 2006*), suggesting that emergence represents a graded and prolonged return to the normal awake state rather than a simple reversal of anesthesia induction. It is still unclear what ongoing brain dynamics contribute to altered sensory processing during emergence from anesthesia.

Here, we use two independent datasets – intracranial recordings from patients emerging from anesthesia after surgery, and high-density EEG recordings from a study of emergence in healthy volunteers under controlled laboratory conditions – to provide a multiscale analysis of neural dynamics during emergence from anesthesia. By defining the trajectory of changes in ongoing neural dynamics and sensory evoked responses during the process of emergence, we identify a new transitional brain state that occurs just before emergence from anesthesia. This state is marked by stimulus-evoked cortical down states that resemble the K-complexes which are hallmarks of stage two non-rapid eye movement (N2) sleep. However, its spontaneous dynamics qualitatively differ from sleep. We show that this state occurs primarily in the minutes prior to awakening, identifying a novel transitional brain state that is selective to anesthetic emergence.

## Results

We analyzed intracranial recordings from 12 patients (13 sessions) with intractable epilepsy during emergence from propofol general anesthesia. Subjects were implanted with subdural electrocorticography (ECoG) and/or penetrating depth electrodes (1095 total electrodes). Emergence recordings took place immediately after completion of clinically indicated surgery to implant intracranial electrodes. Recordings began during maintenance of anesthesia through the clinical infusion of propofol (*Figure 1a*), and continued after the infusion was stopped as the patient emerged from anesthesia and regained consciousness. In 8 of these subjects, recordings were also obtained during a gradual anesthetic induction when patients returned for a second surgery 1–3 weeks later. We presented auditory stimuli every ~3–6 s throughout the emergence period, allowing us to assess cortical

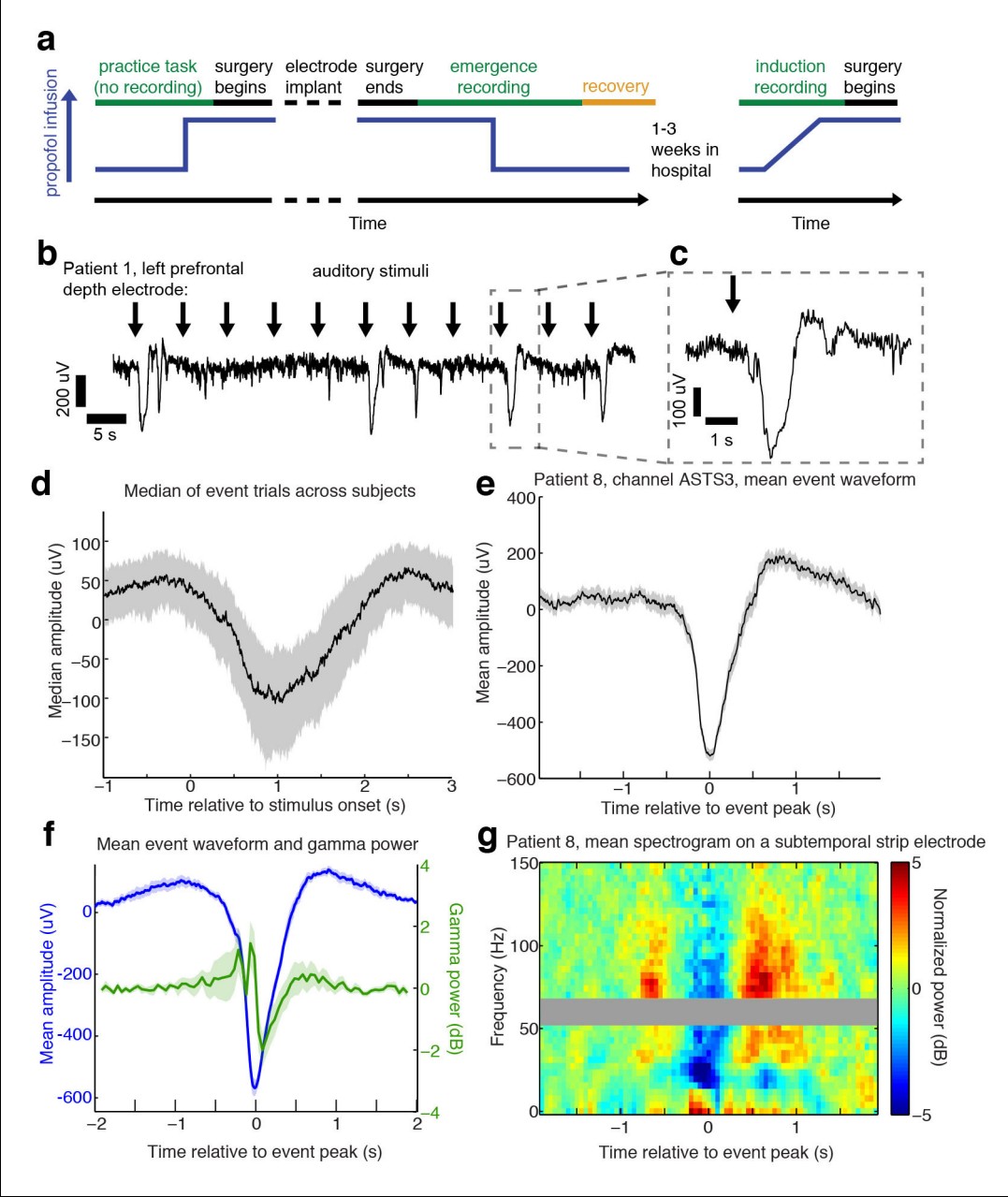

**Figure 1.** Auditory stimuli can evoke large, slow, asymmetric events during emergence from general anesthesia. (A) Schematic of experimental procedures for intracranial study. (B) Example trace from a single subject shows that a subset of auditory trials induce large evoked potentials. Data were lowpass filtered at 40 Hz and downsampled to 100 Hz for display. (C) Zoomed-in view of one event, with gray box highlighting region in zoomed-out view. (D) Median of events within 2 s of stimulus onset across all subjects, including all channels with at least five event trials (n = 190 channels). Shaded region is quartiles. Sign has been flipped to be negative across all channels. (E) Mean waveform in channel with the most events for an example subject, aligned to the event peak (n = 38 events). Shaded region is standard error. (F) Mean waveform aligned to peak across all subjects and mean gamma power during the event (n = 13 sessions, 13 channels, 1339 events). Shaded region is standard error. (G) Normalized spectrogram of high frequencies in same subject and channel as panel E, shows large drop in gamma power during events detected in individual subjects. Gray bar placed around frequency range with 60 Hz line noise.
DOI: https://doi.org/10.7554/eLife.33250.003

The following source data and figure supplement are available for figure 1:

**Source data 1.** Mean processed LPs for each intracranial subject during emergence.
DOI: https://doi.org/10.7554/eLife.33250.005

*Figure 1 continued on next page*

*Figure 1 continued*

**Figure supplement 1.** Schematic of datasets used.
DOI: https://doi.org/10.7554/eLife.33250.004

reactivity to sensory stimuli as drug levels declined. Auditory stimuli consisted of either click trains at 40 Hz in one ear and 84 Hz in the other (2 s duration), non-verbal sounds, or spoken words (see Materials and methods).

## Auditory stimuli can induce large-amplitude evoked potentials during emergence

During emergence from general anesthesia, we observed that in a subset of trials, auditory stimuli elicited a large (>100 µV) and slow (duration >1 s) evoked potential (*Figure 1b,c*) across many electrodes. We developed an automatic detection algorithm to identify these events, which we termed large potentials (LP). LPs were defined as events of >400 µV amplitude lasting >400 ms (see Materials and methods for additional details). We chose these thresholds to conservatively detect only large events while ignoring small or ambiguous LP-like events. 16% of electrodes (n = 1095 electrodes) exhibited at least five events using this detector. This number included electrodes from every patient, as at least two electrodes with $\geq$ 5 LPs were detected in each emergence session. To characterize the relationship between these events and the auditory stimulation, we analyzed all trials on which an LP occurred within two seconds of stimulus onset. The mean stimulus-triggered event on each electrode (*Figure 1d*) had a median peak amplitude of 236 µV (quartile range (QR): 183–295), a value that was lower than the detection threshold due to averaging together events with slightly different peak times. The peak of the mean stimulus-triggered event occurred 1.01 s (QR:0.7–1.38) after stimulus onset, and lasted 0.28 s (full-width at half max; QR: 0.05–0.47), a waveform that was far slower and larger in amplitude than typical auditory-evoked responses in the awake state. The LPs thus rank among the largest electrophysiological signals observed in human cortex, indicating synchronization of electrical signaling among a substantial fraction of the local neuronal population.

The average stimulus-aligned waveform across patients can be temporally blurred due to differences in timing across subjects and electrode locations. To more precisely assess the amplitude and waveform of these events, we selected the electrode with the most events in each subject, and analyzed the mean waveform of all detected events aligned to their peak. The peak-aligned events on these electrodes were larger (median amplitude = 550 µV) and had an asymmetric morphology (*Figure 1c, d, e*), with a sharper onset than offset (mean rise = 165 ms, mean fall = 285 ms, 95% confidence interval (CI) for difference=[84 156] ms, bootstrap; p=0.0002, Wilcoxon signed-rank test) and large post-peak rebound. Aligning to stimulus onset thus confirmed these events were auditory-evoked, whereas analyses aligning to the peak demonstrated that the waveform of the events was large and asymmetric, with substantial variability in exact time-to-peak.

The large, slow, and asymmetric waveform of the LPs resembles K-complexes (KCs), a characteristic electrophysiological graphoelement that occurs spontaneously or following sensory stimulation during stage two non-rapid eye movement (NREM) sleep (*Loomis et al., 1938*; *Colrain, 2005*; *Halász, 2016*). The KC corresponds to a cortical DOWN state (*Cash et al., 2009*), in which local neuronal firing is suppressed. To test whether LPs mark a similar cortical dynamic, we analyzed high-frequency power in the LFP, which is correlated with local spike rates (*Ray and Maunsell, 2011*), during all detected LPs. We selected the electrode with the most LPs in each subject and computed the peak-triggered power, and found that LPs correlated with a strong reduction in broadband gamma-range (40–100 Hz) power (−1.29 dB, CI=[−0.4–2.5], bootstrap; p=0.04, Wilcoxon signed-rank test, *Figure 1f,g*), suggesting they too represent a DOWN state with suppression of neural activity. This peak-locked analysis included both stimulus-evoked and spontaneous events. A substantial proportion (28%) of detected LPs appeared to occur spontaneously, as they were not preceded by an experimental stimulus within 2 s, although other auditory input present in the clinical environment may have contributed to their generation. When the spectral analysis was instead performed relative to the onset of the auditory stimulus, including only trials where LPs appeared within 2 s of a stimulus, we found that this decrease in high-frequency power reached a minimum at 1.3 s post-stimulus, suggesting that the auditory-evoked potentials were also associated with prolonged

suppression of neuronal activity. This slow timecourse is also similar to the timing of auditory-evoked KCs during sleep (*Colrain et al., 1999*).

## Large evoked potentials involve a spatially distributed frontotemporal network

Intracranial recordings provide precise, millimetre-scale spatial resolution, enabling mapping of the cortical sources of LPs. We measured the amplitude of the mean stimulus-evoked response across all electrodes, on trials that evoked an LP in at least one electrode. We aligned mean responses to stimulus onset, to allow consistent comparisons across channels that could exhibit different peak times. Most subjects exhibited LPs on multiple electrodes, with amplitude of the evoked potential varying widely across regions (*Figure 2a*). However, many electrodes exhibited no sign of an LP

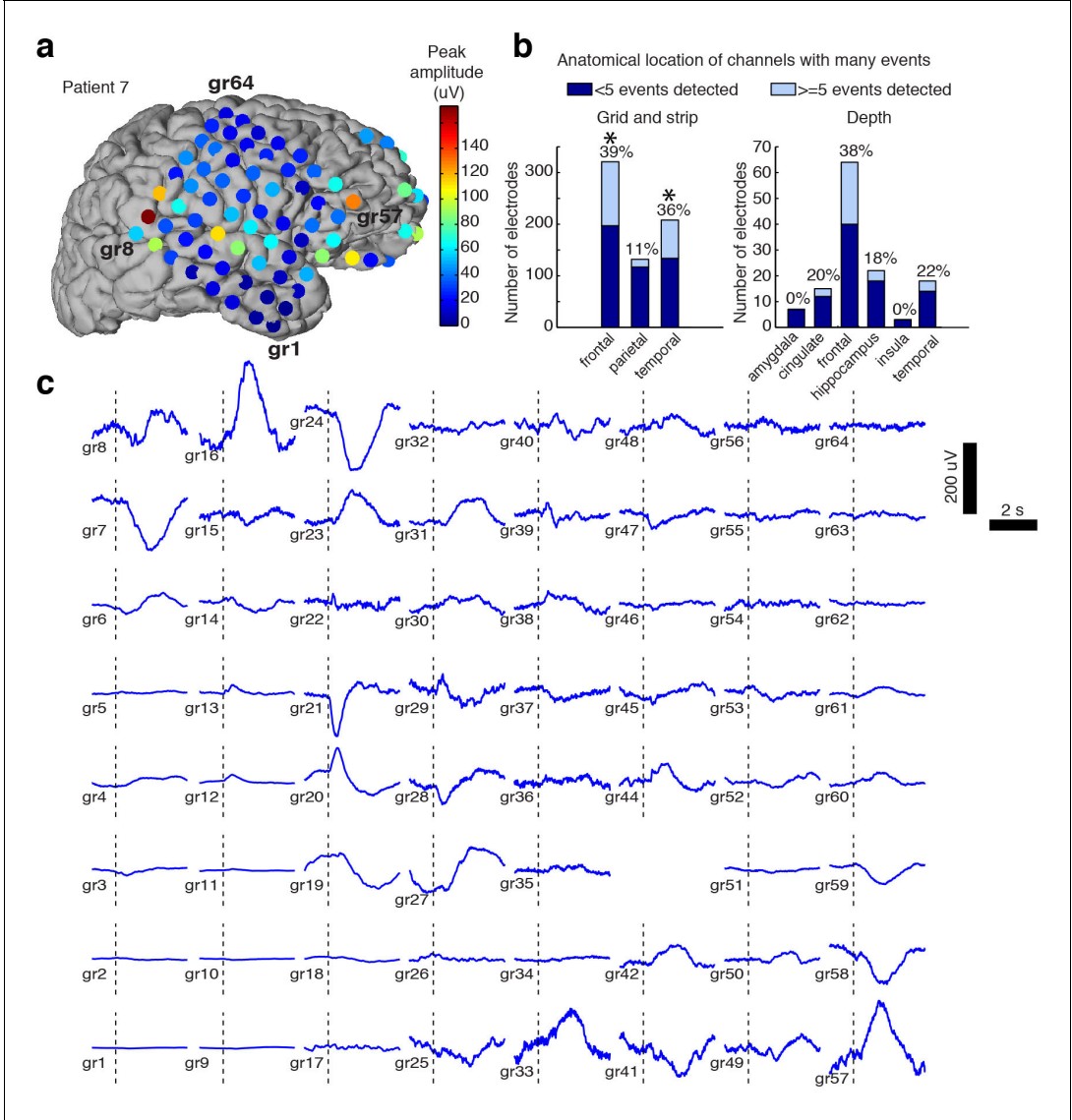

**Figure 2.** Spatial distribution of the large evoked potential. (A) Example surface reconstruction of a single subject, where color indicates absolute value of the amplitude of the mean potential over all 'event trials'. (B) Number of electrodes exhibiting LP occurrence across regions (excluding white matter and regions with <= 5 electrodes). Stars indicate regions with significantly higher proportions of high-LP channels than the mean rate (alpha = 0.05, Bonferroni corrected). (C) Timecourse of the mean stimulus-evoked potential across a grid of electrodes in Patient seven shows that evoked potentials can occur on a large number of channels (black dashed line shows time of stimulus onset).
DOI: https://doi.org/10.7554/eLife.33250.006

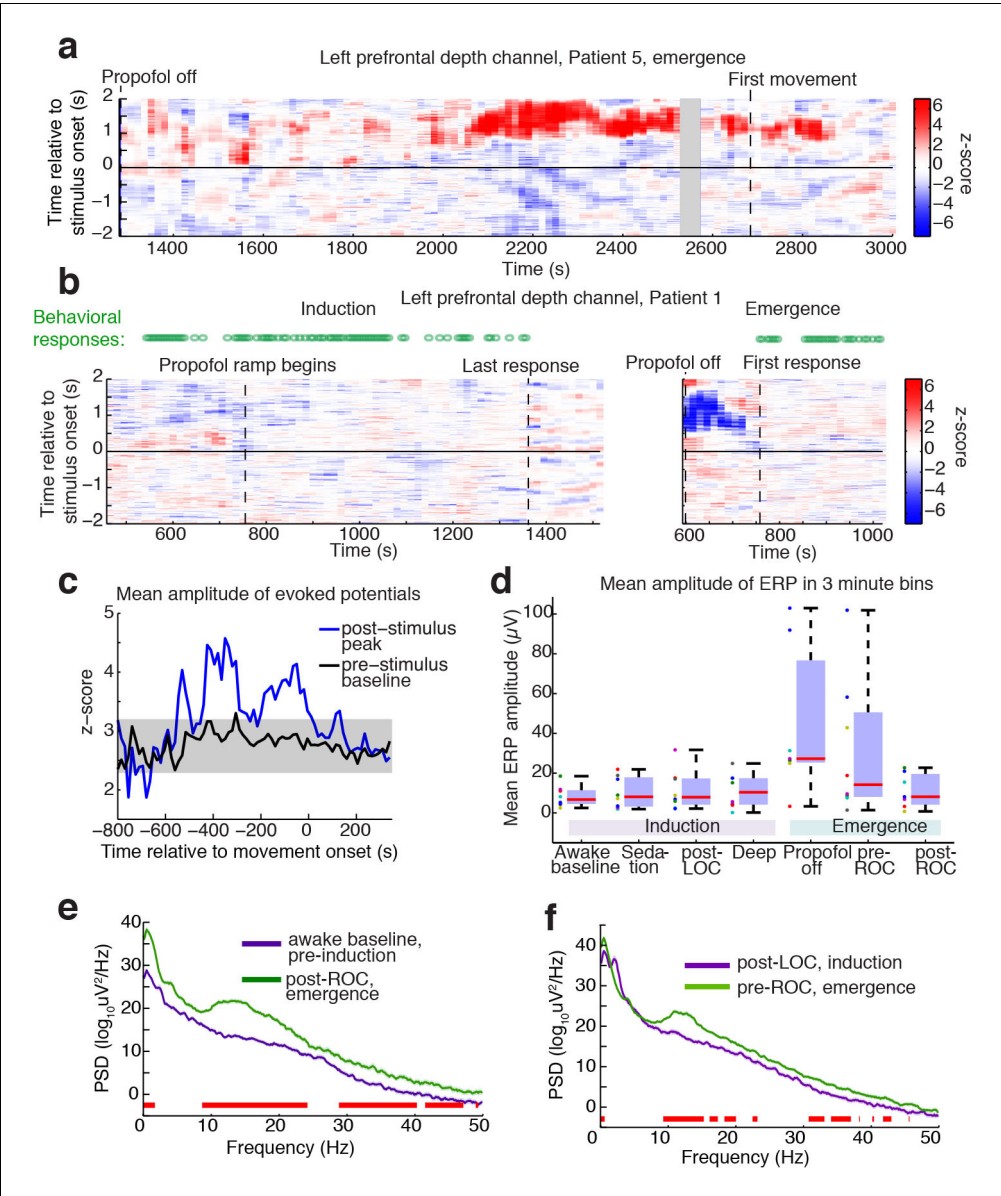

**Figure 3.** Large evoked potentials occur during a brief transitional state. (**A**) Evoked potentials in a patient with a long emergence recording, shows that LPs appear as the propofol concentration declines, and then subside shortly after the patient's first spontaneous movement (seen at ~2200–2900 s). Z-scored ERPs averaged in sliding window of 60 s every 15 s. Gray shading covers window with insufficient (<8) events for averaging. (**B**) Normalized evoked potentials in a patient with both an emergence and an induction recording. The pattern is asymmetric, with stimulus-locked LPs occurring only during emergence. Z-score shows mean ERP normalized to the pre-stimulus baseline in each time window. This patient was under light anesthesia at the end of surgery and LPs appeared even before the propofol infusion was turned off. (**C**) Amplitude of the peak ERP across all subjects, locked to ROC (movement onset) and normalized to pre-stimulus baseline. Evoked potential amplitude across all subjects peaks in the 400 s prior to ROC, and then returns to baseline after ROC, indicating that LPs mostly occur in the minutes preceding ROC. As a control, the peak pre-stimulus baseline z-score across subjects is plotted in black, with gray shading indicating its mean value ±3*st.dev. over time. (**D**) Boxplot of absolute value of mean ERP amplitude at 0.5–1 s post-stimulus in the eight subjects with both induction and emergence recordings. ERPs are small at baseline, sedation, and post-LOC. They are largest in the bin after propofol is turned off and before ROC. (**E**) Mean spectra across patients within the same 3 min time bins, red bars indicate frequency bands with significant difference (p<0.05, bootstrap). The post-ROC state has greater low-frequency (<2 Hz), alpha/beta (~10–24 Hz) and gamma (~30–50 Hz) power than the awake pre-anesthesia baseline (n = 7 subjects). (**F**) Same,

*Figure 3 continued on next page*

*Figure 3 continued*

demonstrates a broadband increase in power above 10 Hz in the emerging state, relative to immediately after LOC during induction (n = 8 subjects). Red bars indicate significant differences at p<0.05 (bootstrap).

DOI: https://doi.org/10.7554/eLife.33250.007

The following source data is available for figure 3:

**Source data 1.** Mean amplitudes of the ERP for each intracranial subject across conditions.

DOI: https://doi.org/10.7554/eLife.33250.008

despite showing ongoing local electrophysiological activity, indicating that these were not global cortical events. The percentage of grid and strip electrodes with at least five detected LPs was highest in frontal and temporal cortex (39% of frontal electrodes, 36% of temporal electrodes, *Figure 2b*), which had significantly higher proportions than the mean rate (26%, CI=[24 29], p<0.05, Bonferroni corrected binomial test). Fewer parietal electrodes exhibited detectable LPs (11%). We also found that LPs were recorded on 35 of the 129 depth electrodes placed in gray matter (27%), including on deep contacts placed in hippocampus. The peak timing and morphology of the evoked potential varied across space within individuals (*Figure 2c*). Overall, our intracranial recordings suggest that LPs were restricted to a specific frontotemporal network of cortical regions rather than a globally coherent slow wave.

## Sensory-evoked LPs occur during a time-limited transitional state

To determine the timecourse of the stimulus-evoked LPs, we computed sliding window measures of the mean evoked response over time, including all trials, on the electrode for each subject that exhibited the most LPs. The LPs were primarily observed after propofol was turned off but before the patient exhibited signs of recovery (*Figure 3a,b*). This effect was seen in the mean amplitude of the ERP over time: the normalized ERP amplitude increased across subjects in the ~300 s prior to the first behavioral sign of emergence, and subsided again thereafter (*Figure 3c*). In the eight patients who had recordings in both induction and emergence, we analyzed the mean ERP amplitude relative to behavioral state changes and found that the LPs occurred predominantly during emergence, particularly in the pre-return of consciousness (pre-ROC) period, and not during induction (*Figure 3d*). Since our induction used a gradual infusion (*Figure 1a*), patients were guaranteed to pass through a plasma concentration level during induction that matched their level at emergence, demonstrating this transient state was selective to the process of emergence rather than only a particular dosage level.

To test what ongoing dynamics accompanied this transitional LP state, we analyzed spectral content within each epoch. We found that the dynamics during emergence were substantially different from induction, exhibiting significantly greater low-frequency (<2 Hz) and alpha power even after awakening (*Figure 3e*). Comparing the three minutes immediately after behaviorally defined loss of consciousness (LOC) during induction, and the three minutes immediately prior to return of consciousness (ROC), a smaller but otherwise similar power difference was evident (*Figure 3f*).

## Evoked LPs detected in scalp EEG reveal asymmetric induction and emergence dynamics

While the intracranial recordings suggested asymmetry between induction and emergence, due to time constraints in the operating room we were not able to measure intracranially over prolonged periods. To test these dynamics in a more controlled setting and in a population of healthy subjects, we next analyzed scalp EEG data recorded during a stepped infusion of propofol in healthy volunteers (*Purdon et al., 2013*; *Mukamel et al., 2014*), during presentation of auditory stimuli that were click trains, words, and the subject's own name. This stepped infusion protocol induced slow changes in propofol concentration and behavioral responses (*Figure 4a,b*). The steady-state auditory evoked potential to the auditory click train stimuli also declined, quantified as the induced power at 40 Hz, corresponding to the click frequency (*Figure 4c*). To confirm this decrease was selective to the auditory-evoked band rather than broadband, we also analyzed power at a 'control' frequency of 22 Hz (i.e., not the stimulus frequency) and found no change. We next examined the traces and

found that large evoked potentials were clearly visible during emergence after large-amplitude slow oscillations subside (*Figure 4d,e,f*).

To apply the same LP detector, we focused on a frontal EEG electrode, as frontal electrodes had high LP rates in our intracranial data (*Figure 2b*) and did not exhibit the large auditory-evoked potentials of the temporal electrodes. We detected stimulus-evoked LPs (peak >7 s.d., *Figure 4*) during emergence from propofol anesthesia in 4 out of 10 subjects, despite the lower spatial resolution of the scalp recordings (*Figure 4e,f,g*). If the detection threshold was lowered (peak >5 s.d.), we could also observe brief traces of similar events in the induction period in 3 of the 10 subjects. However, these periods were brief and infrequent (*Figure 4e*), suggesting that this brain state occurs primarily (but not exclusively) during emergence (*Figure 4—figure supplement 1*). While we detected these events in a frontal electrode, LP events were observed broadly across the scalp (*Figure 4—figure supplement 1*), consistent with the widespread spatial profiles we observed in the intracranial data. These results in healthy volunteers confirmed that the LPs were not related to epileptic events in the patients. Furthermore, they show that LPs occurred primarily during emergence (*Figure 4h,i*, fig. supp. 1) even in these experiments with a prolonged induction period, lasting more than twice as long as the emergence period.

In these subjects, the LPs were also found to be stimulus-selective: they occurred preferentially in response to the sound of words and names, and did not occur following click-train stimuli (*Figure 4h*, *Figure 4—figure supplement 1*). In contrast, no such stimulus selectivity was observed in the intracranial patients, as each stimulus type could elicit LPs (in the channel with the most events in each subject, LPs occurred within 2 s of 21% of word stimuli; 20% of sounds stimuli; 22% of click-train stimuli). A key difference between these two datasets was the relative frequency of the name and word stimulus categories, which were infrequent (20% names/words, 80% clicks) in the scalp data but were evenly distributed in the intracranial data (30% words, 40% clicks, 30% sounds). The increased saliency of an infrequent stimulus may thus increase the probability of an LP, similar to reports for KCs during sleep (*Colrain et al., 1999*).

We observed LPs for a prolonged period that could extend after the initial ROC in the scalp EEG dataset (*Figure 4h,i*, *Figure 4—figure supplement 1*), whereas LPs were only present before ROC in the intracranial dataset (*Figure 3d*, *Figure 4—figure supplement 1*). This difference likely reflects the differences in arousal state across these two datasets: in the intracranial study, the drug was completely shut off and patients emerged rapidly as drug levels monotonically decreased. In contrast, in the scalp EEG dataset, the propofol levels were lowered in a gradual, stepped fashion (*Figure 4a*), leading to a prolonged emergence period over tens of minutes. These large LPs therefore may be present not only in the minutes prior to any sign of ROC, but may continue through emergence until a relatively heightened arousal state is reached.

## Evoked responses strongly resemble spontaneous K-complexes during sleep

Although the LPs shared some properties with the spontaneous KCs that occur during N2 sleep, the propofol emergence period could be expected to also exhibit significant differences from natural sleep. To test the similarity between events during sleep and during emergence, we obtained intracranial recordings during sleep from a subset of the patients (n = 3 patients). To compare the LPs detected during propofol with the spontaneous KCs during sleep, we first verified that the automatic detection algorithm could identify events in the sleep datasets. We found that 64% of manually identified KCs were also flagged by the automatic detector, suggesting this approach could be used to quantitatively compare the two phenomena within this patient cohort (although the high number of misses, 36%, suggests it should not be employed as a KC detector for more general purposes).

The LPs recorded during emergence and the sleep KCs shared an overall profile of large (>100 µV) and slow (~1 s) waveforms (*Figure 5a*). No significant difference in median amplitude was seen between the propofol and sleep datasets (grouped median in sleep = 644 µV (CI=[630 673]), propofol = 647 µV (CI=[623 665]), bootstrap; p=0.59, Wilcoxon rank-sum test), and the distributions of event amplitudes shared a high degree of overlap (*Figure 5b*).

The spatial distribution of sleep KCs also appeared very similar to that seen during emergence from propofol anesthesia (*Figure 5c,d*). To test this spatial similarity, we computed the mean event waveform across all electrodes, triggered on the peak of each event detected in a single electrode with a large number of events in both the sleep and propofol recordings. We found that the mean

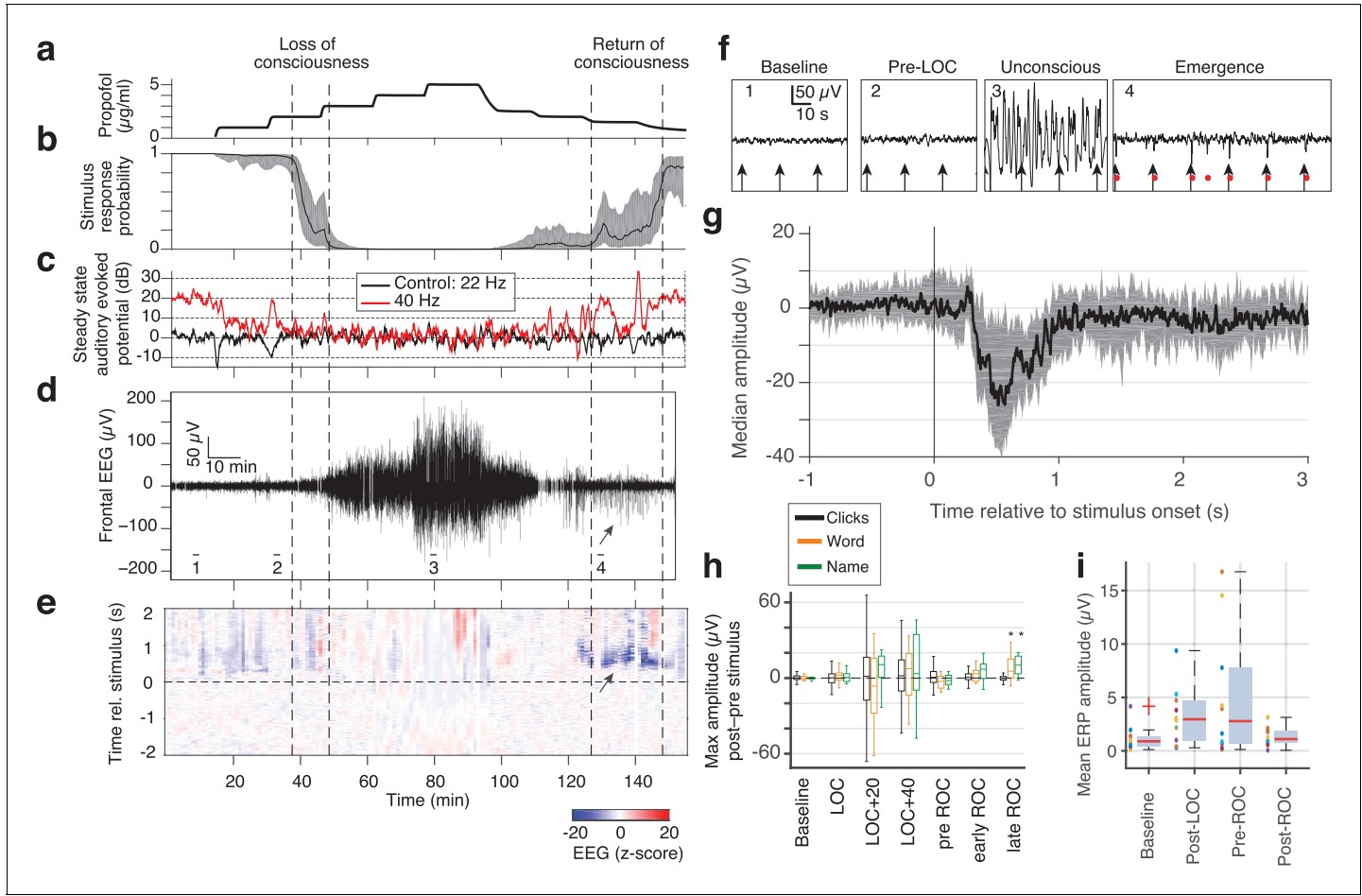

**Figure 4.** Large evoked potentials can be detected during emergence in scalp EEG recordings. (**A**) Experimental design for scalp EEG volunteer study: target propofol plasma concentration increased and decreased as a controlled step function. (**B**) Behavioural response rate in one subject shows times of loss of consciousness (LOC) and recovery of consciousness (ROC). (**C**) Steady state auditory evoked potential (SSAEP) at lateral electrodes to the click train stimulus (repetition rate 40 Hz, red) declines during propofol general anesthesia; the control frequency (22 Hz) shows no SSAEP. (**D**) EEG at a central electrode over the entire recording shows high-amplitude, low-frequency power during the anesthetized period. Prior to ROC (around 120 min, marker 4), large unipolar negative potentials are observed (arrow). (**E**) Stimulus-triggered average exhibits large, slow potentials prior to return to the awake state, but not during induction or during maintenance. (**F**) Example records from the epochs marked in panel D show that sensory-evoked LPs in the scalp EEG occur during emergence (4), but not pre-anesthesia baseline (1), pre-LOC (2) or profound unconsciousness (3). (**G**) Median of events in a central channel occurring during ROC following click stimuli. Shaded region is interquartile range. (**H**) Timecourse of ERP amplitudes in the same individual subject: boxplot indicates variability of mean ERP amplitude across individual trials. LPs selectively appear during emergence, and are evoked by the infrequent word and name stimuli (orange, green), but not by the frequent noise bursts (black). Stars indicate significance at p<0.01 with Wilcoxon signed-rank test. This subject exhibits a slow emergence with LPs appearing around and after the initial onset of ROC. (**I**) Boxplot of absolute value of mean ERP amplitude across all subjects estimated within 5 min windows during each epoch. Colored points show individual subjects.

DOI: https://doi.org/10.7554/eLife.33250.009

The following source data and figure supplement are available for figure 4:

**Source data 1.** Mean ERP for each EEG subject (baseline).
DOI: https://doi.org/10.7554/eLife.33250.011
**Source data 2.** Mean ERP for each EEG subject (post LOC).
DOI: https://doi.org/10.7554/eLife.33250.012
**Source data 3.** Mean ERP for each EEG subject (post ROC).
DOI: https://doi.org/10.7554/eLife.33250.013
**Source data 4.** Mean ERP for each EEG subject (pre ROC).
DOI: https://doi.org/10.7554/eLife.33250.014
**Figure supplement 1.** Evoked potentials in scalp EEG dataset.
DOI: https://doi.org/10.7554/eLife.33250.010

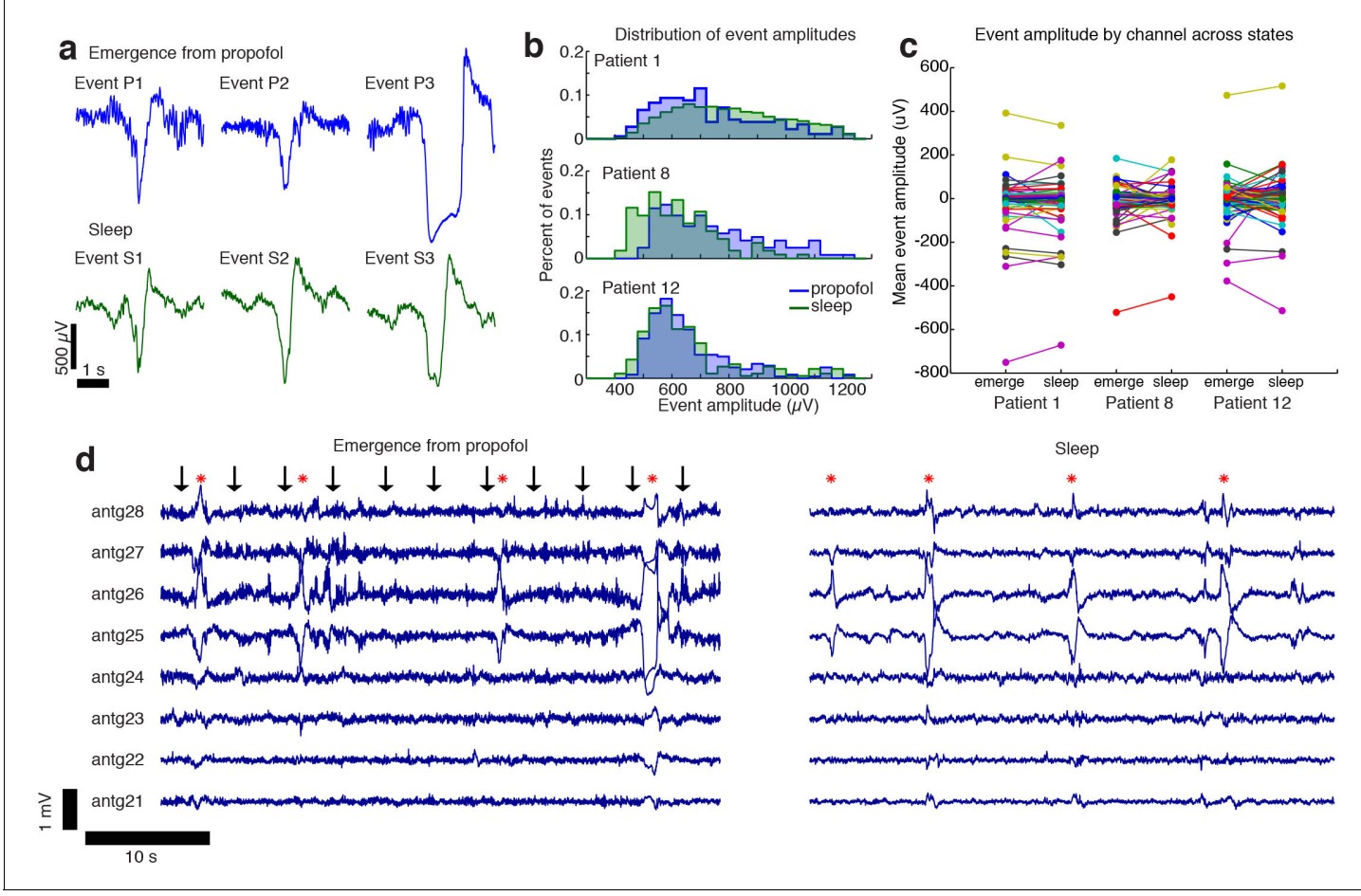

**Figure 5.** The events observed during propofol emergence resemble K-complexes seen during natural sleep. (A) Example traces of automatically detected events on a single channel during propofol emergence and during sleep in the same subject. (B) Distribution of absolute value of event amplitudes in emergence and in sleep. Waveform distributions are very similar across the two datasets. (C) The amplitude of the mean event on individual channels is highly consistent across sleep and emergence; large values are the channels with large LPs. Channels with large LPs consistently also show large KCs; channels with no LPs (near-zero amplitudes) do not have large KCs during sleep. In each patient, the spatial profile of events is highly consistent across sleep and emergence. (D) Timeseries across eight grid channels during emergence and during sleep demonstrates that the LPs typically occur on the same channels and with the same polarity as the KCs seen in natural sleep. Red stars indicate events detected on multiple channels; black arrows show timing of auditory stimuli.

DOI: https://doi.org/10.7554/eLife.33250.015

The following source data and figure supplement are available for figure 5:

**Source data 1.** Channelwise peak amplitudes (Patient 1).
DOI: https://doi.org/10.7554/eLife.33250.017
**Source data 2.** Channelwise peak amplitudes (Patient 8).
DOI: https://doi.org/10.7554/eLife.33250.018
**Source data 3.** Channelwise peak amplitudes (Patient 12).
DOI: https://doi.org/10.7554/eLife.33250.019
**Figure supplement 1.** Correlated spatial properties of events across emergence and sleep.
DOI: https://doi.org/10.7554/eLife.33250.016

event amplitude across electrodes was significantly correlated when comparing sleep and propofol (R = 0.92,0.62,0.71, for the three patients, *Figure 5c*, *Figure 5—figure supplement 1*), meaning that electrodes with large KCs were likely to also exhibit large LPs during emergence. Overall, the shared waveform, spatial distribution, and timing of these events suggest that the LPs observed during propofol emergence may engage the circuit mechanisms that generate KCs during natural sleep.

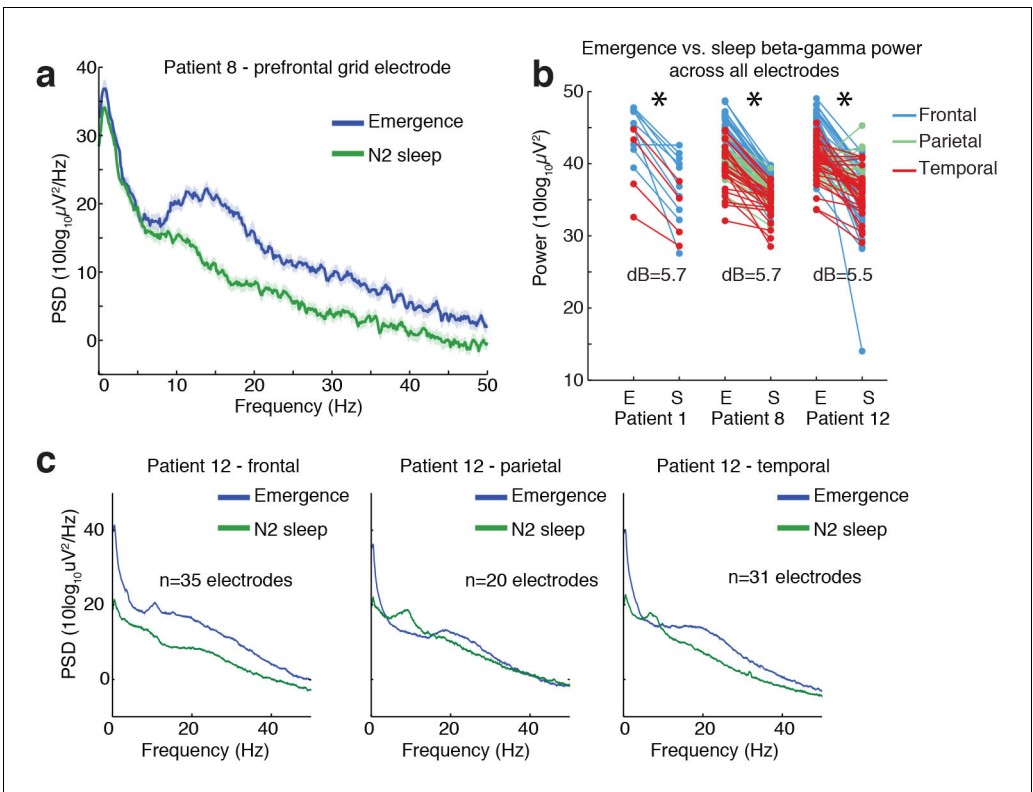

**Figure 6.** Ongoing electrophysiological dynamics differ between sleep and emergence. (**A**) Example spectrum from a single channel: emergence is associated with stronger broadband power at frequencies >10 Hz than N2 sleep. (**B**) Difference in 10–40 Hz power across all electrodes in emergence (E) vs. N2 sleep (S), categorized by anatomical location: power is significantly lower across all electrodes during sleep in each subject (p<0.001, signed-rank test). (**C**) Spatial profile of dynamics in a single patient categorized by spatial location: while alpha and spindle dynamics vary spatially, 10–40 Hz power is elevated during emergence across a broad cortical region.
DOI: https://doi.org/10.7554/eLife.33250.020

Given the resemblance of the LPs to KCs, we next tested whether ongoing spectral dynamics within the LP period resembled N2 sleep. We computed the power spectrum during a manually selected period exhibiting LPs during emergence, and compared these to segments of sleep recordings manually identified as N2 sleep. We found substantial differences in these spectra, with propofol emergence exhibiting more power across a broad frequency range of 10 to 40 Hz throughout all recorded cortical regions (*Figure 6*, median difference = 5.6 dB, CI=[3.9 6.1], bootstrap; p<0.001 in each subject, Wilcoxon signed-rank test). In addition, the sleep spectra exhibited clear spindle power (10–14 Hz) peaks across cortical regions, whereas the emergence spectra exhibited either no peak or a spatially restricted frontal alpha (~10 Hz) peak characteristic of deep propofol anesthesia (*Figure 6c*). These results demonstrate that while some common neurophysiological events can be observed in stage two sleep and in this transient emergence period, emergence is a distinct brain state that is not identical to sleep.

## Discussion

Using both intracranial ECoG and scalp EEG recordings, we found that emergence from general anesthesia is accompanied by a transient state in which auditory stimuli can evoke large potentials (LPs) corresponding to all-or-none cortical suppressions lasting several hundred milliseconds. LPs strongly resemble the K-complexes observed in N2 sleep, although the neural dynamics of emergence from general anesthesia nevertheless represent a distinct state. This state appeared primarily during emergence and foreshadowed the return of behavioral responsiveness, suggesting it represents a distinct brain state through which patients transition as they recover consciousness. Our data

indicate that the brain's response to propofol is hysteretic, such that the current state is determined not only by the drug concentration but also by the recent history of the brain's activation.

The brain state we observed appears to be distinct from the sedated state experienced by patients during slow anesthetic induction, as it was exclusively observed during emergence and not induction of general anesthesia in the intracranial recordings. A small number of LPs were detected during induction of anesthesia in a subset of subjects over the course of extremely long (>1 hr) inductions in the scalp EEG dataset, but these were rare and vastly outnumbered by the more frequent LPs occurring during emergence. In addition, a previous intracranial study of slow (~1 hr) inductions of propofol general anesthesia did not report analogous events (*Nourski et al., 2017*), suggesting this phenomenon is primarily a signature of emergence. We also found that this transitional state is not identical to sleep: comparing neural dynamics during sleep and emergence from general anesthesia in the same subjects identified substantial differences in the power spectrum. The frontal alpha rhythm characteristic of propofol anesthesia is still present during emergence (*Feshchenko et al., 2004*; *Murphy et al., 2011*; *Purdon et al., 2013*), but not during N2 sleep, indicating these are distinct brain states. Spontaneous alpha rhythms during propofol are thought to be generated by increased inhibitory tone in thalamic circuits, causing an intrinsic ~10 Hz dynamic to emerge (*Ching et al., 2010*). These alpha rhythms are still present during the LPs, suggesting the thalamus may be exhibiting an altered excitatory/inhibitory balance as compared to sleep. However, despite the difference in spontaneous dynamics, the LP events themselves share many common properties with sleep, exhibiting highly similar waveforms and spatial profiles. In addition, LPs occurred at higher rates in response to more salient stimuli. A similar effect has also been found in sleep, as salient stimuli (such as rare stimuli or the subject's own name) produce larger KC peaks during sleep (*Colrain et al., 1999*; *Perrin et al., 1999*). These events may therefore reflect an analogous effect of arousing stimuli in sleep and emergence, which could conceivably be related to some similarity in circuit state, such as ongoing tonic vs. bursting dynamics in thalamus. The common morphology of the LPs we observe during emergence and the KCs characteristic of sleep suggest that similar circuit mechanisms are engaged by auditory stimuli despite differences in the ongoing spontaneous dynamics.

There is evidence that neuromodulatory arousal systems mediate emergence from general anesthesia, distinct from induction. Disruption of orexinergic signaling increases the time required for emergence from anesthesia, but does not change the dose-response sensitivity for induction (*Kelz et al., 2008*). Coherent alpha (8–12 Hz) and delta (1–4 Hz) oscillations develop rapidly and pervasively across medial prefrontal cortex and thalamus at loss of consciousness induced by propofol, and likely mediate the functional disruption of these areas, contributing to the state of unconsciousness (*Flores et al., 2017*). During emergence, these oscillations dissipate in a sequence distinct from induction, beginning with superficial cortical layers and medial and intralaminar thalamic nuclei, following known cortical and thalamic projection patterns for dopaminergic and cholinergic signaling (*Flores et al., 2017*). Neuromodulatory activity during emergence could therefore create unique cortical and thalamic circuit states that enable LP responses to sensory stimulation. Given the similarity between LPs under anesthesia and sleep K-complexes, similar mechanisms might also play a role in modulating levels of arousal during sleep.

The LPs we observe are qualitatively different from the ongoing slow oscillations that occur during deep anesthesia (*Steriade et al., 1993*; *Breshears et al., 2010*; *Murphy et al., 2011*; *Lewis et al., 2012*). LPs occur after the ongoing slow oscillation has largely subsided and reflect an isolated cortical DOWN state elicited by auditory stimulation, rather than a rhythmic cortical dynamic. However, the occurrence of LPs increases power in the same low-frequency bands of the spectrum that are occupied by the slow oscillation. Future studies may therefore need to take care that their analyses differentiate between these two distinct states, as increased low-frequency power may indicate isolated LP occurrence and foreshadow awakening, and will be important to distinguish from the slow oscillations of deep anesthesia.

While the LPs were strikingly large, they may have been obscured in previous studies due to the brief and transient nature of the state in which they occur. In addition, we observed substantial heterogeneity across patients in terms of the frequency and timing of the LPs. In the intracranial data this heterogeneity may be partially explained by variation in electrode location, the duration and complexity of the surgery, and dosage of clinical medications administered to each patient. In the scalp EEG data, however, drug levels were controlled and no surgery was performed, yet

heterogeneity across subjects was still present. This heterogeneity is also consistent with clinical observations, as patients are much more variable in how long they take to emerge than they are in induction. Following anesthetic emergence, patients exhibit variable levels of arousal, with some patients taking hours to return to alertness (*Larsen et al., 2000*). While animal studies have reported stereotyped transitions between states (*Hudson et al., 2014*), possibly due to increased experimental control and genetic similarity between individual animals, human studies have suggested that individuals may exhibit different trajectories during emergence from anesthesia (*Hight et al., 2014*), and undergo different transitions between distinct, potentially sleep-like states (*Chander et al., 2014*). This variability may reflect individual differences in arousal regulatory circuits or even in drug diffusion rates across the brain. It may also be that some patients pass through the transient stage too quickly for it to be identified using our analyses. Another possibility is individual physiological differences, such as receptor density or vascular properties, could modulate the relative rate of drug clearance in cortex and subcortex, and that only some individuals may experience this state. However, since these events were detected in all the intracranial patients we studied, it may be that the transient state occurs in most patients but is more challenging to detect in scalp EEG due to blurring of signals measured at the scalp. In addition, the healthy volunteers received a smaller total amount of propofol than the clinical patients, and may therefore have been more likely to emerge too rapidly to detect this brief state.

The precise circuit mechanisms that generate the LP phenomenon are not clear, and will be challenging to identify with certainty using data from human subjects. However, we suggest that the sleep K-complex may share some mechanistic parallels with the LPs observed here. The KC is an isolated cortical DOWN state (*Cash et al., 2009*) and is likely to also involve thalamic circuits (*Jahnke et al., 2012*; *Mak-McCully et al., 2014*). While previous animal studies have identified spontaneous KCs during maintenance of ketamine-xylazine anesthesia (*Amzica and Steriade, 1998a*), these occurred as part of an ongoing slow oscillation rather than the isolated auditory-evoked events seen here and during N2 sleep. Moreover, those events were not selective to anesthetic emergence, suggesting they represent a different phenomenon. Stimulus-evoked potentials in animal studies have primarily reported stimulus-evoked responses with a faster timecourse than the LPs reported here (*Amzica and Steriade, 1998b*), perhaps reflecting a different phenomenon during relatively stable states of anesthesia in most animal studies, compared with dynamic changes during awakening. Future animal studies should therefore track the gradual process of emergence to identify the mechanisms of the isolated LPs identified here. One possible mechanism is that increased thalamic activation leads to strong stimulation of the thalamic reticular nucleus (TRN), leading to a thalamic and subsequent cortical suppression. This theory would be consistent with animal studies that have induced slow waves through stimulation of TRN and suppression of thalamocortical neurons (*Lewis et al., 2015*), and with human imaging studies demonstrating that emergence is associated with increased activity in subcortical arousal structures such as thalamus (*Långsjö et al., 2012*). Alternatively, it may be that an inhibitory shift in the excitatory/inhibitory balance in cortex leads to a local profound suppression in response to sensory input, generating a local LP that can then spread across cortex or through corticothalamic projections. Future studies could explore these theories further through causal manipulations of cortical and thalamic activity during a gradual emergence process.

These future investigations could also address some limitations of the current study. As intracranial electrodes are placed solely based on clinical need, we did not obtain whole-brain coverage, and had no thalamic recordings. Animal studies could investigate more systematically the spatial profile of the observed LPs. In addition, due to the nature of our experiment taking place in the operating room, we were constrained in timing and could not record throughout a continuous induction, maintenance, and emergence. In addition, the induction and emergence recordings were not counterbalanced in time due to the ordering of implant and explant procedures. They could potentially exhibit small changes in electrode position and signal-to-noise-ratio. While our data suggest no major difference in recording quality that could explain the striking LPs we observe, and we observe the LPs in scalp EEG as well despite opposite temporal ordering, more subtle phenomena could depend on differences in the recordings across these sessions. Highly controlled volunteer studies, as in our scalp data, will therefore be useful counterparts to any future intracranial investigations of these phenomena. Finally, our patient sample was small due to the rare nature of these recordings, and therefore we could not examine how the heterogeneity of LP dynamics might relate to

emergence time or other clinical outcomes. Gathering datasets in larger patient cohorts would be very valuable for investigating how these dynamics can inform patient monitoring and predict functional outcomes. In particular, the LP events could potentially be used to monitor depth of anesthesia or predict when a patient will emerge, or they may be found to relate to emergence-related clinical outcomes, such as delirium. Future clinical studies would be highly beneficial for investigating these questions.

In summary, we identified a transient brain state that occurs asymmetrically during emergence from general anesthesia. While deep states of anesthesia have been well characterized and exhibit stereotyped electrophysiological signatures, tracking transitions between states demonstrates the existence of transient and heterogeneous dynamics that occur selectively in the minutes before emergence. This state engages similar sensory-evoked circuit dynamics as in sleep, suggesting the brain may sometimes experience a sleep-like sensory blockade before recovering from general anesthesia.

## Materials and methods

### Clinical setting

Written informed consent was obtained from all patients and experimental procedures were approved by the Massachusetts General Hospital/Brigham and Women's Hospital Institutional Review Board. The enrolled patients had medically intractable epilepsy and underwent surgery to implant intracranial electrodes for clinical monitoring purposes (*Figure 1—figure supplement 1*). The location and number of electrodes implanted was determined by clinical criteria without regard to this study. Recordings were performed in the hospital operating room as the patients emerged from propofol general anesthesia. Recordings began after surgery was completed and while the clinical infusion of propofol was still running, and continued throughout the period after the infusion was stopped and patients emerged from anesthesia, until patients had to be disconnected for transport outside of the operating room. No seizures were recorded in these data. Patients received the typical clinical regimen of medication throughout the surgery (including paralytics and analgesics), and in most cases the maintenance infusion also included remifentanil. We acquired intracranial recordings from 15 patients during emergence. Data from two patients were excluded due to poor recording quality, and data from one patient was excluded due to failure of the auditory stimulus equipment. A second emergence recording was acquired from one patient with electrodes implanted in different locations, and this recording was treated as another subject in the analysis (total analyzed = 13 sessions, drawn from 12 individuals, five female, mean age 34.5 years, range 21–48 years). In four sessions, patients had only depth electrodes, and in nine sessions they had both subdural grid/strip and depth electrodes. Eight of these patients were also studied during gradual induction of general anesthesia when they returned 1–3 weeks later to undergo electrode removal surgery. In the induction recording, propofol was infused gradually using STANPUMP software with a target plasma concentration rising linearly over 10 min to a maximum of 6 µg/mL (*Schnider et al., 1999*).

### Behavioral task – Intracranial recordings

Auditory stimuli were presented every 3.5–4.5 s with uniform temporal jittering (11 sessions) or every 6 s (two sessions) using EPrime software and air-tube earphones to avoid stimulus-related artifacts in the electrophysiology data. Stimuli consisted of either a click train with a frequency of 40 Hz in one ear and 84 Hz in the other, lasting 2 s; a non-verbal sound (e.g. door closing, alarm); or a spoken word. Stimulus types were pseudorandomized throughout the presentation. Words and sounds were of neutral or negative affect; these distinctions were not analyzed in detail here. During the induction of anesthesia prior to the start of the surgery, patients listened to the stimuli and were asked to press a button to indicate whether the stimulus was a word. During emergence, stimulus presentation began near the time that the propofol infusion was stopped, and continued until the patient became responsive. The total presentation duration was 20 min, and if the patient had not yet emerged at that time then the presentation was restarted. Only two patients began performing the task at emergence. Due to this behavioral observation, clinical staff also periodically (approximately every ~1–2 min) asked subjects to open their eyes. Return of behavioral responsiveness was

marked manually using two definitions: the first spontaneous movement observed by research staff (labeled 'First movement'), and the time at which patients began responding to verbal requests to open their eyes or move their hands (labeled 'First response', defined as return of consciousness (ROC)). In 8 of these patients, the same behavioral task was used during induction of anesthesia 1–3 weeks later when patients returned for surgery to remove the intracranial electrodes. The task began 4 min prior to the start of the gradual propofol infusion and continued for 4 min after the target plasma level reached its maximum level.

## Intracranial electrophysiology data

During anesthetic induction and emergence, intracranial recordings were acquired from depth and/ or subdural grid and strip electrodes, with placement selected solely by clinical staff for clinical purposes. Recordings were acquired with an XLTEK acquisition system at a 2000 Hz sampling rate. Bad electrodes were manually identified and excluded from further analysis. Depth and strip electrodes were re-referenced to a bipolar montage in which an adjacent contact was subtracted from each channel. Grid electrodes were referenced to a Laplacian montage by subtracting the mean of the immediately neighboring electrodes. Data were detrended, lowpass filtered below 200 Hz, downsampled to 500 Hz, and highpass filtered above 0.16 Hz.

## Automated event detection

The automated detector was designed to conservatively select events, missing some events but also reducing false positives. Since occasional large artifacts interfered with event detection, automatic detection of spontaneous events was restricted to the longest manually identified continuous segment with acceptable recording quality. All other timepoints were excluded from the automatic detection window. This approach was chosen due to the nature of the intracranial recording: we began recording as soon as possible, but clinical interaction with the patient at the beginning and end of the experiment, as well as connecting and disconnecting electrodes, led to very large artifacts at these timepoints whereas we obtained a long, stable recording during the emergence process. The median duration of this segment across patients was 650 s (inter-quartile range: 580–1410 s). This long segment typically still included some periods with noise, which were rejected automatically in further analyses as described below. Data were first filtered between 0.2–4 Hz. All positive and negative peaks with an amplitude of at least 400 μV were identified. The duration of this peak, defined as the amount of time spent over a threshold of 40 μV, was required to be at least 400 ms. Peaks with amplitude greater than 1200 μV were discarded as artifact, and events occurring within 500 ms of a previous event were discarded. All events within a single electrode were required to have the same polarity, selected as whichever polarity was most frequent across all automatically detected events, since the referencing montage allowed potentials to be either negative- or positive-going depending on local polarity and electrode positioning.

## Event-locked analysis

Trials with a range (peak-trough difference) exceeding 1500 μV were discarded as artifact. Event trials were defined as those trials with an automatically detected event occurring within 2 s of stimulus onset. The mean of all event trials was computed for each electrode that had at least five event trials. Because different electrodes had different polarities, the sign for negative-going electrodes was flipped. The median and quartiles were then computed across the pool of all electrode waveforms (restricted to electrodes with at least five event trials) and all 12 patients. Analyses of individual waveforms (e.g. *Figure 2c*) selected the electrode with the most detected events in each patient. Rise times and fall times were computed on the mean waveform for each selected electrode, by calculating the amount of time it took to rise and fall from a threshold of 200 μV to the peak of the mean event waveform. Bootstrapped 95% confidence intervals were calculated by resampling across subjects with replacement 1000 times, and reporting the 2.5th and 97.5th percentile of the resulting distribution.

## Spectral analysis

Spectrograms were computed using the electrode with the most LPs in each subject. Triggered spectrograms were computed relative to the peak of the LP waveform selected by the automatic

detector. Spectral analysis was performed using multi-taper estimation (Chronux, http://chronux.org, [Bokil et al., 2010]). The analysis used three tapers and a sliding window of 200 ms duration every 50 ms. Spectrograms were normalized within frequencies to the mean power at that frequency between $[-2-1]$ s prior to the peak. Broadband gamma power was computed by taking the mean power between 40 and 100 Hz, relative to the mean gamma power in the $[-2-1]$ s window. Statistical analyses of gamma power were performed on the mean gamma power in the 300 milliseconds post-peak using the Wilcoxon signed-rank test. Spectra for ongoing spontaneous dynamics (Figure 3) used six 30 s epochs within a continuous 3 min time window, using 19 tapers. Spectra were downsampled by a factor of 4 for display. Statistical comparison between time windows was performed by a hierarchical bootstrap resampling procedure: (1) resample subjects; (2) resample epochs within subjects; (3) compute the mean spectrum for each time window on the resampled time windows; (4) calculate the difference between the two spectra. This procedure was repeated 1000 times to obtain 1000 bootstrap estimates of the difference in the spectra; differences outside the [2.5,97.5] percentile for more contiguous frequency points than the spectral resolution of the multitaper estimate were labelled as significant and marked in red. One subject was excluded from the post-ROC vs. awake baseline comparison because electrode quality became too poor (s.d. >500 µV) after the patient emerged due to motion artifacts. Shaded error bars in the plot were computed in Chronux using jackknife estimation.

## Spatial analysis of evoked potentials

Because the automatic detector imposes an artificial threshold on amplitude for events, the spatial analysis was performed on the stimulus-evoked potential over all electrodes. This analysis included only trials that were identified as generating an LP on at least one electrode, and excluded any trials with amplitude above 1500 µV as noise. The peak amplitude of the mean evoked potential in each electrode was plotted in color on a 3D reconstruction of the cortical surface generated using Freesurfer (Fischl, 2012) and with grid and strip electrode coordinates registered to the surface of the brain (Dykstra et al., 2012). To categorize the spatial location of electrodes, the nearest anatomical label from the Freesurfer automatic subcortical segmentation or cortical parcellation (Destrieux et al., 2010) was assigned. Electrodes identified as being in white matter and electrodes in regions with fewer than five contacts (e.g., putamen, occipital cortex) were excluded from the spatial analysis. Statistical testing of which of the nine regions had significantly high proportions of electrodes with >5 LPs was performed with a binomial test, comparing each region to the mean across regions, with a Bonferroni correction for multiple comparisons across regions. Displayed grid timecourses are lowpass filtered below 30 Hz and downsampled to 100 Hz for display.

## Timecourse analysis

Sliding window plots over induction and emergence were calculated by averaging all trials within a window of 60 s sliding every 15 s. For z-score analysis (Figure 3c), the peak amplitude of the ERP was normalized to the standard deviation of the 1.5 s pre-stimulus, across each 60 s window, and the plots display the resulting z-scores. Calculations were only included when at least eight stimuli occurred within the window. When analyzing mean evoked amplitude across time windows (Figure 3d), a 3 min period for each window was defined, and the mean evoked response was computed. The mean amplitude in the 0.5–1.5 post-stimulus window was then computed for each subject. As before, one subject was excluded from the post-ROC condition because electrodes began to be disconnected and recording quality was not usable.

## Sleep-intracranial comparison

Recordings of natural sleep were obtained for three of the intracranial recording patients during their hospital stay (after the emergence recording and prior to the induction recording). An experienced neurophysiologist (G.P.) scored the sleep data and manually labelled the onset and offset times of a subset of clearly visible KCs in the intracranial recordings for initial validation of the approach. Sleep data was acquired on a clinical system with a sampling rate of either 500 or 512 Hz. To match the propofol recording, the same reference electrodes were used for each electrode as in the emergence dataset, and then all electrodes were filtered between 0.16 Hz and 200 Hz. Any electrodes where the same reference electrodes were not available in both datasets were excluded. For

analysis of median peak amplitude in individual events, electrodes with at least four events in each dataset were included. The histogram reflects all detected events on these electrodes, whereas the statistical test drew the same number of events from both the sleep and the propofol datasets for each subject. For bootstrap confidence interval estimation, data across subjects were pooled due to the small number of patients, and the bootstrap drew from datapoints pooled across the three patients. For comparison, within-subject statistics are also presented. To compare the spatial distribution of events across both datasets, event times were selected from a single electrode with at least 20 events in both datasets, and then the peak-triggered waveform across all electrodes was computed using these selected times. The mean value of the peak-triggered waveform between 100 ms pre-peak and 100 ms post-peak was calculated, and this mean event value was then compared across electrodes. Spectra were compared by identifying four 30 s windows of clean recordings with high LP rates in the emergence dataset, and randomly selecting four 30 s consecutive windows of N2 in the sleep dataset. Spectra were computed using Chronux with 19 tapers, downsampled by a factor of 4 for display, and error bars were computed with the jackknife method at $p < 0.05$.

### Scalp EEG dataset

Scalp EEG analysis used data that was previously published (*Purdon et al., 2013*; *Mukamel et al., 2014*) with full details provided in those publications. Briefly, healthy volunteers underwent monitoring with 64-channel EEG during a slow infusion of propofol, targeting a stepped increase from 0 to 5 µg/mL plasma concentration over one hour, and then a stepped decrease until the subjects recovered consciousness. Stimuli consisted of click trains (2 s duration), words, or the subject's own name, with stimulus type pseudorandomized throughout the experiment. 80% of the stimuli were click trains, 10% were words, and 10% were names. The LP analysis used a single frontal EEG electrode. For each stimulus presentation, we subtracted the mean and divided by the standard deviation during the 2 s pre-stimulus period. We then computed the maximum stimulus-evoked amplitude during the 1 s following stimulus presentation, and averaged these over 1 min windows.

## Acknowledgements

We would like to thank Lisa Feldman Barrett for advising on stimulus design and Aaron Sampson for assistance with experiments. This work was funded by a Junior Fellowship from the Harvard Society of Fellows, and NIH grants K99-MH111748, R00-NS080911, DP2-OD006454, S10-RR023401, R01-NS062092, R01AG056015, P01GM118269, and R01-EB009282.

## Additional information

### Competing interests

Laura D Lewis: Co-author of pending patents on EEG monitoring for general anesthesia (US20140316217A1). Emery N Brown, Patrick L Purdon: Co-author of pending patents on EEG monitoring for general anesthesia (US20140316217A1, WO2015108908A3). Eran A Mukamel: Co-author of pending patents on EEG monitoring for general anesthesia (WO2015108908A3). The other authors declare that no competing interests exist.

### Funding

| Funder | Grant reference number | Author |
| --- | --- | --- |
| National Institutes of Health | K99-MH111748 | Laura D Lewis<br>Sydney S Cash<br>Emery N Brown<br>Eran A Mukamel<br>Patrick L Purdon |
| Harvard University | Society of Fellows | Laura D Lewis |

| National Institutes of Health | DP2-OD006454 | Laura D Lewis<br>Sydney S Cash<br>Emery N Brown<br>Eran A Mukamel<br>Patrick L Purdon |
|---|---|---|
| National Institutes of Health | S10-RR023401 | Laura D Lewis<br>Sydney S Cash<br>Emery N Brown<br>Eran A Mukamel<br>Patrick L Purdon |
| National Institutes of Health | R01-EB009282 | Laura D Lewis<br>Emery N Brown<br>Eran A Mukamel<br>Sydney S Cash<br>Patrick L Purdon |
| National Institutes of Health | R01-NS062092 | Laura D Lewis<br>Emery N Brown<br>Eran A Mukamel<br>Sydney S Cash<br>Patrick L Purdon |
| National Institutes of Health | P01-GM118269 | Emery N Brown |
| National Institutes of Health | R00-NS080911 | Eran A Mukamel |
| National Institutes of Health | R01-AG056015 | Patrick L Purdon |

The funders had no role in study design, data collection and interpretation, or the decision to submit the work for publication.

### Author contributions

Laura D Lewis, Conceptualization, Formal analysis, Funding acquisition, Investigation, Methodology, Writing—original draft, Project administration, Writing—review and editing; Giovanni Piantoni, Resources, Formal analysis, Methodology, Writing—review and editing; Robert A Peterfreund, Emad N Eskandar, Priscilla Grace Harrell, Linda S Aglio, Resources, Investigation; Oluwaseun Akeju, Resources, Investigation, Writing—review and editing; Sydney S Cash, Conceptualization, Resources, Investigation, Methodology, Project administration, Writing—review and editing; Emery N Brown, Conceptualization, Funding acquisition, Methodology, Writing—review and editing; Eran A Mukamel, Conceptualization, Formal analysis, Funding acquisition, Investigation, Methodology, Writing—original draft, Writing—review and editing; Patrick L Purdon, Conceptualization, Resources, Funding acquisition, Investigation, Methodology, Writing—review and editing

### Author ORCIDs

Laura D Lewis http://orcid.org/0000-0002-4003-0277
Giovanni Piantoni http://orcid.org/0000-0002-5308-926X
Oluwaseun Akeju https://orcid.org/0000-0002-6740-1250
Eran A Mukamel http://orcid.org/0000-0003-3203-9535

### Ethics

Human subjects: Written informed consent was obtained from all patients and experimental procedures were approved by the Massachusetts General Hospital/Brigham and Women's Hospital Institutional Review Board (protocol numbers 2010P000093 and 2005P001549).

### Decision letter and Author response

Decision letter https://doi.org/10.7554/eLife.33250.023
Author response https://doi.org/10.7554/eLife.33250.024

## Additional files

### Supplementary files

• Transparent reporting form
DOI: https://doi.org/10.7554/eLife.33250.021

### Data availability

The processed LPs for each intracranial subject during emergence (Figure 1), the mean amplitudes of the ERP for each subject across conditions (Figure 3), and the channelwise peak amplitudes for individual subjects (Figure 5) are provided. For Figure 2, the underlying data are fully represented in the figure. For the scalp EEG dataset, the processed ERPs for the displayed subject with LPs and the mean amplitudes of the ERP across subjects (Figure 4) have been uploaded. Software for event detection is available at https://github.com/lauralewis/emergenceLPs (copy archived at https://github.com/elifesciences-publications/emergenceLPs).

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
