## [Decision Letter]

Thank you for submitting your article "A transient cortical state with sleep-like sensory responses precedes emergence from general anesthesia in humans" for consideration by *eLife*. Your article has been favorably evaluated by Michael Frank as the Senior Editor and Saskia Haegens as the Reviewing Editor. The following individuals involved in review of your submission have agreed to reveal their identity: Eelke Spaak and Jan Claassen.

The reviewers have discussed the reviews with one another and the Reviewing Editor has drafted this decision to help you prepare a revised submission.

Summary:

Lewis et al. recorded intracranial ECoG data from a group of surgical epilepsy patients during the induction of and emergence from propofol anesthesia. They identified a so-called "large potential" (LP) in the electrophysiological data, which they observed specifically during the emergence stage. LPs were observed primarily in response to auditory stimulation, but also spontaneously. LPs share clear spatial and temporal characteristics with K-complexes, as observed during stage 2 non-REM (N2) sleep. However, the spectral characteristics of emergence from propofol anesthesia and N2 sleep are quite distinct. Therefore, the authors identify emergence from anesthesia as a distinct brain state. Some of the findings were reproduced recording EEG in a separate sample of healthy controls undergoing controlled propofol anesthesia.

Essential revisions:

The reviewers believe this work interesting and the manuscript well-written; however, they raise a number of concerns that must be adequately addressed before the paper can be considered for publication in *eLife*.

First, some aspects of the experimental paradigm remain unclear:

1) What was the duration of a single auditory stimulus? Also, please define the nature of the stimuli early on in the manuscript; now it is only explained during the results of the EEG participants. Furthermore, the paradigms differed between cases. Was the sequence of different stimuli (like words and tones) randomized or kept constant?

2) It is entirely unclear why the 22-Hz stimulus in the EEG study should be considered a "control" stimulus (as opposed to the "experimental" 40-Hz stimulus). Also, the use of steady-state evoked potentials in general is not introduced anywhere in the text.

Furthermore, several questions were raised regarding the data acquisition:

3) Of course by necessity, the recordings in patients were not counterbalanced in time, with up to 3 weeks in between, during which changes can have occurred (e.g. electrode position, impedance, SNR, etc.) which are otherwise unrelated to the experimental conditions but may have impacted the results. It would strengthen their claims if the authors could show some other measure that did not differ between these two time points, e.g. evoked response to the stimuli during wakefulness (or any other way to take away concerns about time-order effects here). Related, did the authors record any seizures pre or during the case?

4) What was the impact of location of intracranial recording electrodes? Interpreting the significance of the LP observations is challenging with a small diverse patient cohort and an arbitrarily defined detector.

5) Regarding the scalp EEG dataset, why was a single frontal electrode chosen for the LP analysis, as presumably most recordings in epilepsy patients actually used temporal lobe coverage? If the authors later conclude that frontal networks underlie the down states indicated by large LPs and decreased gamma why did they then not analyze the most frontal intracranial electrodes?

Critically, several questions and concerns were raised regarding the analysis procedure:

6) What was the variability of the length of recording segments for automatic event detection? Why not take the entire recording and just reject artifact-contaminated parts?

7) Please quantify the heterogeneity in LPs (onset/amplitude) locked to stimulus onset. Especially given the large fraction of LPs that did not appear stimulus-locked at all, it would be good to demonstrate that the majority of LPs *is* indeed related to the auditory stimuli. Related to this, the analyses and figures flip back and forth between time axes locked to event peaks (e.g. Figure 1E/F/G) and stimulus onsets (e.g. Figure 2). Please stick to one definition of the time axes, or clearly motivate different choices.

8) Did lowering the threshold lead to observation of LP during emergence in more of the EEG subjects (subsection “Evoked LPs detected in scalp EEG reveal asymmetric induction and emergence dynamics”, first paragraph) ? Also, please provide illustrations of individual and mean ERPs/LPs in the EEG study as well (i.e. analogous to Figures 1D/E/F), separately for different anesthesia epochs. Furthermore, source level analysis might help recover LPs in more EEG subjects, considering it would improve the spatial resolution.

9) Although the authors demonstrate that LPs are not present during either induction of anesthesia or alert wakefulness, they provide no evidence that LPs are absent during anesthesia itself. Please quantify. (In fact, looking at Figure 3A, it appears that there are some relatively high-amplitude ERPs before the 'propofol off' marker.)

10) The authors report that their detector picked up 64% of manually identified K-complexes, but how many were missed? After running this on three patients they should not claim that this may be used as an automatic detector.

11) The correlations between K-complex and LP amplitudes across channels (Figure 5D) appear to be largely driven by outliers for two out of the three patients (8 and 12). Please replot the scatter plots with the most extreme outliers removed, to get a more detailed view of the central cloud of points.

Finally, it would be helpful if the authors could expand on the interpretation and implication of their findings, highlighting the importance and novelty of their study:

12) Is there any clinical relevance, can this be used as some sort of indicator? Was there a correlation with the LP measure (difference across subjects) and other perhaps behavioral measures? Did length/number/strength of LP occurrence correlate with speed of anesthesia recovery, for instance?

13) The authors suggest that this is completely novel, however, they do not cite rather older papers that described the same phenomenon during sleep (Perrin et al., 1999: "A differential brain response to the subject's own name persists during sleep").

14) What is the relation between on the one hand alpha and LP, and on the other hand the inhibitory nature of the LPs (considering that alpha is also thought to have an inhibitory function). Is there a direct mechanistic link between the two? What is the causal order of events here?

[Editors' note: further revisions were requested prior to acceptance, as described below.]

Thank you for resubmitting your work entitled "A transient cortical state with sleep-like sensory responses precedes emergence from general anesthesia in humans" for further consideration at *eLife*. Your revised article has been favorably evaluated by Michael Frank (Senior Editor), a Reviewing Editor, and one reviewer.

The manuscript has been substantially improved but there are some remaining issues that need to be addressed before acceptance, as outlined below:

1) In Figure 4H, it appears as though LPs are most prominent in scalp EEG post-ROC (ROC+20), whereas in Figure 4I, the peri-ROC bar appears the highest. For the intracranial data, it was clearly the pre-ROC period that shows the strongest LPs (Figure 3D). The authors don't reflect on this difference.

2) It should be noted that Figures 4H/4I/3D all appear to quantify the same thing (LP magnitude of some sort), but the exact quantity which the authors chose to plot differs (max/mean/median amplitude). As I pointed out in my original review, it would be good to unify units as much as possible among plots.

---

## [Author Response]

Essential revisions:The reviewers believe this work interesting and the manuscript well-written; however, they raise a number of concerns that must be adequately addressed before the paper can be considered for publication in eLife.First, some aspects of the experimental paradigm remain unclear:1) What was the duration of a single auditory stimulus? Also, please define the nature of the stimuli early on in the manuscript; now it is only explained during the results of the EEG participants. Furthermore, the paradigms differed between cases. Was the sequence of different stimuli (like words and tones) randomized or kept constant?

We have now added more description of the stimulus paradigm to the Results (first paragraph and subsection “Evoked LPs detected in scalp EEG reveal asymmetric induction and emergence dynamics”, first paragraph). The click train stimuli used in both the intracranial and EEG paradigms were of 2 s duration (we have added this to Materials and methods,). For the other categories of auditory stimuli (names, words, sounds), the duration varied for each stimulus since the stimuli were heterogeneous (e.g. different word lengths). The sequence of stimuli was pseudorandomized with no pre-specified order of presentation, we have also added this to the Materials and methods (subsections “Behavioral task – Intracranial recordings” and “Scalp EEG dataset”).

2) It is entirely unclear why the 22-Hz stimulus in the EEG study should be considered a "control" stimulus (as opposed to the "experimental" 40-Hz stimulus). Also, the use of steady-state evoked potentials in general is not introduced anywhere in the text.

We apologize for this confusion. There was no 22 Hz stimulus; the stimuli were only 40 Hz and 84 Hz. The triggered power at 22 Hz is shown as a control analysis to demonstrate that the SSAEP we observe is specific to the frequency of stimulation (i.e. that only the induced frequency, and not baseline power, changes). We have added this explanation to the Results (subsection “Evoked LPs detected in scalp EEG reveal asymmetric induction and emergence dynamics”, first paragraph).

Furthermore, several questions were raised regarding the data acquisition:3) Of course by necessity, the recordings in patients were not counterbalanced in time, with up to 3 weeks in between, during which changes can have occurred (e.g. electrode position, impedance, SNR, etc.) which are otherwise unrelated to the experimental conditions but may have impacted the results. It would strengthen their claims if the authors could show some other measure that did not differ between these two time points, e.g. evoked response to the stimuli during wakefulness (or any other way to take away concerns about time-order effects here). Related, did the authors record any seizures pre or during the case?

We did not record any seizures in these data, which we have now added to the Materials and methods (subsection “Clinical setting”). It is challenging to directly compare these recordings since it takes upwards of an hour for surgical-level propofol to completely clear from the system, whereas electrodes had to be disconnected rapidly after emergence so that patients could be transported out of the OR. Patients are therefore typically awake but not fully alert in our post-ROC condition, preventing direct comparison with the awake baseline prior to induction, since arousal state is lower. Due to this, no exactly equivalent brain states are studied across the two conditions. However, overall low-frequency power is very similar across induction and emergence recordings at similar dosage levels (Figure 3F), arguing against any gross change in recording quality large enough to explain these >100 µV stimulus-evoked differences. In addition, the strong agreement between our emergence session LPs and the KCs from the sleep sessions acquired later in the week (Figure 5), with tightly matched amplitudes and spatial properties, suggests no major change in recording quality has occurred over the course of a few days. We agree with reviewers that small shifts in electrode position and SNR may occur across these sessions, and have added a description of this limitation to the Discussion (seventh paragraph), but our analyses in Figures 3 and 5 demonstrate these effects are unlikely to be able to explain the striking stimulus-locked phenomena we observe.

In addition, our scalp EEG data confirms this phenomenon cannot be due to order effects: the appearance of LPs occurs preferentially during emergence in our scalp EEG data as well, even though temporal order is reversed in this experiment. We have added this issue to the Discussion (seventh paragraph), mentioning this limitation, and that we do not observe any major changes in recording quality (at least not on a large enough scale to explain the very large LP differences we see).

4) What was the impact of location of intracranial recording electrodes? Interpreting the significance of the LP observations is challenging with a small diverse patient cohort and an arbitrarily defined detector.

Although the placement of electrodes in each patient was heterogeneous and determined by clinical criteria, most patients did have a broad range of brain regions covered including substantial parts of the temporal, frontal and parietal lobes (total = 1,095 electrodes in the 13 sessions). Figure 2 shows the impact of recording location in one example patient (Figure 2A, C) and across our entire cohort (Figure 2B), demonstrating that the LP rate does vary by spatial location, and is highest in frontal and temporal locations. In our EEG subjects we recorded 64-channel EEG across the entire scalp and analyzed the spatial pattern of the LPs, identifying widespread appearance of these signals (Figure 4G). In addition, we also provide trial-averaged responses that do not depend on the detector (Figure 3A, B, C, D), to demonstrate a consistent effect during emergence even when a detector is not used to segment these trials. We have also added this limitation to the Discussion (seventh paragraph), suggesting future studies that could investigate these spatial properties further by including larger patient cohorts or using animal recordings.

5) Regarding the scalp EEG dataset, why was a single frontal electrode chosen for the LP analysis, as presumably most recordings in epilepsy patients actually used temporal lobe coverage? If the authors later conclude that frontal networks underlie the down states indicated by large LPs and decreased gamma why did they then not analyze the most frontal intracranial electrodes?

We observed the highest LP rate in our intracranial frontal electrodes (Figure 2B), suggesting this is a prime area for investigating LP phenomena, and our patients generally had extensive frontal and temporal electrodes present. The EEG data do not have sufficient spatial resolution to draw highly localized conclusions about the networks involved in generating LPs. Indeed, we observed that LPs, when they occur, generally are detectable across many scalp EEG electrodes, consistent with the fact that these events were intracranially detected over more than a third of frontal intracranial electrodes. We therefore chose to detect LPs using a single EEG electrode for simplicity. The choice of a frontal electrode was also made to avoid confounding the LPs with stimulus-evoked activity (auditory potentials) we observed more strongly at temporal electrodes. We have added this reasoning to the Results (subsection “Evoked LPs detected in scalp EEG reveal asymmetric induction and emergence dynamics”, second paragraph).

Critically, several questions and concerns were raised regarding the analysis procedure:6) What was the variability of the length of recording segments for automatic event detection? Why not take the entire recording and just reject artifact-contaminated parts?

We have added an explanation to this to the subsection “Automated event detection” and have now added quantification of the length and variability of the recording segment. The length is highly variable as patients take very heterogeneous amounts of time to emerge from anesthesia. Our recording segmentation approach was due to the clinical setting of these acquisitions: we segmented out the portion of the recording that was usable, because the beginning and end of the recordings are irrecoverably noisy due to the fact that we are interacting with the patient to connect or disconnect electrodes and stabilize patient position during this time. Manually identifying the stable recording period was therefore helpful for avoiding this excessive noise and periods in which some electrodes may be being adjusted. If this very high-noise period is not excluded, then baseline calculations of standard deviation for automatic noise rejection are biased to be very high (because they include long periods where electrodes are not stable and contain severe noise). Within the segment where we record without interference, we aimed to avoid hand-labeling of individual trials as noisy in order to minimize subjective judgments for what qualified as substantial noise, and instead automatically rejected noisy trials. In general, intracranial ECoG recordings are low-noise, and we used the amplitude threshold described in Materials and methods (see the aforementioned subsection) to automatically reject any trials that were likely to have excessive noise.

7) Please quantify the heterogeneity in LPs (onset/amplitude) locked to stimulus onset. Especially given the large fraction of LPs that did not appear stimulus-locked at all, it would be good to demonstrate that the majority of LPs is indeed related to the auditory stimuli. Related to this, the analyses and figures flip back and forth between time axes locked to event peaks (e.g. Figure 1E/F/G) and stimulus onsets (e.g. Figure 2). Please stick to one definition of the time axes, or clearly motivate different choices.

This issue was due to lack of clarity on our part – this quantification is in Figure 3A, B, C which demonstrates that these LPs are on average locked to the stimulus: this sliding window analysis does not use the automated detector, rather it includes all trials, and demonstrates a consistent large evoked potential is identified during emergence. In addition, Figure 3D demonstrates consistent stimulus-locked potentials (again, not using the detector). We have updated the legend to clarify that this figure is reporting mean evoked potentials without any detector, demonstrating these events are in fact time-locked (albeit with some variability).

Our quantification of heterogeneity in LP amplitude and timing is in the Results subsection “Auditory stimuli can induce large-amplitude evoked potentials during emergence”, first paragraph, reporting median and quartile range of event amplitude, timing, and width, across all subjects.

We think it is valuable to show both kinds of plots (stimulus-locked vs. peak-locked) because they allow visualization of different aspects of the evoked potentials: locking to event peaks shows the typical waveform, whereas locking to stimulus onset shows the typical LP delay relative to the stimulus. As suggested by the reviewers we have now added more explanation of this motivation for the two types of analysis to the Results (subsections “Auditory stimuli can induce large-amplitude evoked potentials during emergence”, second paragraph and “Large evoked potentials involve a spatially distributed frontotemporal network”). We think that showing both types of analyses provides additional useful information and hope this new text will clarify the interpretation of these two differing plots.

8) Did lowering the threshold lead to observation of LP during emergence in more of the EEG subjects (subsection “Evoked LPs detected in scalp EEG reveal asymmetric induction and emergence dynamics”, first paragraph)?

We have now added a new figure showing the effect of altering the threshold (Figure 4—figure supplement 1C). We observed large-amplitude stimulus-evoked potentials (>12 µV) in 4/10 subjects during the pre-ROC period. As suggested by the reviewer, several of the other subjects did have smaller amplitude potentials (<10 µV), but these were in the range of potentials we observed during post-LOC and post-ROC epochs. We therefore conservatively chose to report detection of LPs only in 4/10 subjects since whether these smaller threshold events also corresponded to LPs was not completely unambiguous. In addition, Figure 4I shows the median amplitude of stimulus-evoked potentials in each subject, and this analysis does not impose any threshold, to display the full range of variability across subjects. Overall these analyses confirm that LPs were detected in the ROC period in the scalp EEG subjects.

Also, please provide illustrations of individual and mean ERPs/LPs in the EEG study as well (i.e. analogous to Figures 1D/E/F), separately for different anesthesia epochs.

We have now updated Figure 4 to more closely parallel Figure 1: the individual ERPs are shown in Figure 4F and median ERP is shown in Figure 4G. In addition, we have now plotted ERPs across the separate epochs in the new Figure 4—figure supplement 1B.

Furthermore, source level analysis might help recover LPs in more EEG subjects, considering it would improve the spatial resolution.

Because we had access to high-spatial resolution intracranial recordings that showed these events are both widespread and spatially heterogeneous, we chose not to attempt source level analysis of EEG data, since the correct assumptions for regularization of the spatial inference are less clear in this case of very widespread LPs (detectable across over a third of frontal and temporal intracranial electrodes).

9) Although the authors demonstrate that LPs are not present during either induction of anesthesia or alert wakefulness, they provide no evidence that LPs are absent during anesthesia itself. Please quantify. (In fact, looking at Figure 3A, it appears that there are some relatively high-amplitude ERPs before the 'propofol off' marker.)

We have now added the deep anesthesia condition during the intracranial induction experiment to Figure 3D, demonstrating that LPs are not present at this surgical-level maintenance dose. We do not have an equivalent maintenance-level epoch in the intracranial emergence recording, due to the restriction of recording in the OR environment: we were not able to record during anesthetic maintenance during the electrode implant, instead we connect the electrodes right at the end of the surgery just before anesthesia is turned off, so our emergence recordings do not contain a maintenance period, but our induction recordings do contain the deeply anesthetized maintenance state and our new figure demonstrates that LPs are absent during anesthesia maintenance.

The previous Figure 3A feature commented on was due to high variability in the pre-LOC period when slow oscillations are still present, and reflects an edge effect from our sliding window analysis (which could not smooth as much in the first few time windows due to them containing fewer stimuli and was therefore more easily swayed by noise in this initial period). We realize this was misleading and have updated the figure based on these comments. Now that we have converted to the z-score display as suggested in the comment below, we have removed the edge effects and this time period does not appear (due to it containing <8 stimuli for averaging), which should clarify this figure. Furthermore, these variable spots did not have the same waveform as the LPs, as they contained a strong negative component not seen in the slow, large responses that appear later in emergence (Figure 3a, ~2100-2500 s).

To further address this issue, in our scalp EEG dataset we have now added additional epochs to the analysis as well (Figure 4H) and found no evidence for stimulus-evoked LPs during maintenance of general anesthesia. Note that during general anesthesia many subjects exhibit very strong slow oscillations (0.5-2 Hz). However, as these are not individual events driven by the stimulus, they do not result in the mean stimulus-evoked response that is seen during LPs (Figure 4D, H, Figure 4—figure supplement 1B).

10) The authors report that their detector picked up 64% of manually identified K-complexes, but how many were missed? After running this on three patients they should not claim that this may be used as an automatic detector.

We agree, we have added a statement indicating that our detector is an analysis tool designed specifically for this LP study and is not intended to be a KC detector (subsection “Evoked responses strongly resemble spontaneous K-complexes during sleep”, first paragraph). As 64% of KCs were identified, 36% were missed.

11) The correlations between K-complex and LP amplitudes across channels (Figure 5D) appear to be largely driven by outliers for two out of the three patients (8 and 12). Please replot the scatter plots with the most extreme outliers removed, to get a more detailed view of the central cloud of points.

This issue is due to our lack of clarity in presentation: we would like to clarify that each of the data points in Figure 5 represents the mean response in one electrode. Thus, the large-amplitude data points are not outliers but instead represent the channels with the most robust, largest-amplitude LPs – the highest-signal channels and the ones with the large phenomenon of interest. This figure shows that the same channels which exhibit large-amplitude LPs during emergence from anesthesia show the largest-amplitude K-complex events during sleep. To avoid confusion we have replaced these correlation figures with paired dot plots that more clearly show the magnitude characteristics across patients (new Figure 5C). We also now clarify this aspect of the figure in the legend. The overall finding is that the channels with very large LPs (which resembled outliers in this plot but in fact reflect large evoked responses) are highly consistent across conditions, whereas channels without LPs are also consistently low-amplitude across conditions.

No correlation is expected in the central cloud alone, since this is comprised of channels with near-zero response channels: i.e. the ones that do not actually exhibit LPs, and do not have any correlation in their mean response, since this reflects residual noise in the absence of a stimulus-locked event. Instead, the relevant result is that channels that do have LPs have them in both sleep and anesthesia, whereas channels without LPs have them in neither condition, as illustrated in our new Figure 5C. To provide extra information about the effects we observe, we have also now added bootstrap confidence intervals to complement the Pearson correlation statistical reporting, which reduces sensitivity to outliers, and show that the bootstrap confidence interval is significant within 2/3 individual subjects. We have also added new correlation plots with and without the central cloud as Figure 5—figure supplement 1 – as predicted, correlation increases when the central cloud is removed, as the channels that have LPs are also more correlated across sleep and emergence (since they contain very similar event magnitudes and spatial distributions). Thus, there is reduced correlation within the central cloud without LPs, and consistent matching of LPs within channels that do have large responses.

Finally, it would be helpful if the authors could expand on the interpretation and implication of their findings, highlighting the importance and novelty of their study:12) Is there any clinical relevance, can this be used as some sort of indicator? Was there a correlation with the LP measure (difference across subjects) and other perhaps behavioral measures? Did length/number/strength of LP occurrence correlate with speed of anesthesia recovery, for instance?

We have now added a new Discussion section on clinical relevance of these findings (seventh paragraph). Due to the small patient sample we cannot draw conclusions at this point about correlation with individual patient outcomes, but we now comment on this limitation and suggest future studies that could examine these patterns. In particular, it would be valuable to examine whether LP occurrence can be used to predict impending emergence from anesthesia in a larger clinical study.

13) The authors suggest that this is completely novel, however, they do not cite rather older papers that described the same phenomenon during sleep (Perrin et al., 1999: "A differential brain response to the subject's own name persists during sleep").

We have now added the suggested citation as well as discussion situating our work relative to papers studying KCs in sleep (Discussion, second paragraph).

14) What is the relation between on the one hand alpha and LP, and on the other hand the inhibitory nature of the LPs (considering that alpha is also thought to have an inhibitory function). Is there a direct mechanistic link between the two? What is the causal order of events here?

We have now added discussion about the properties of alpha relative to the LPs that we see. While we can only speculate on the causal relationships in these data, we do observe that spontaneous alpha is high during emergence, but low during sleep, whereas the LPs and KCs are highly similar, leading us to speculate that perhaps that alpha vs. LPs reflect distinct circuit mechanisms (Discussion, second paragraph).

[Editors' note: further revisions were requested prior to acceptance, as described below.]

The manuscript has been substantially improved but there are some remaining issues that need to be addressed before acceptance, as outlined below:1) In Figure 4H, it appears as though LPs are most prominent in scalp EEG post-ROC (ROC+20), whereas in Figure 4I, the peri-ROC bar appears the highest. For the intracranial data, it was clearly the pre-ROC period that shows the strongest LPs (Figure 3D). The authors don't reflect on this difference.

We have now added a section discussing the difference between the intracranial and scalp EEG experiment LP timing (subsection “Evoked LPs detected in scalp EEG reveal asymmetric induction and emergence dynamics”, last paragraph). This pattern is likely due to the differences in experimental design across the two studies: the intracranial study involves a rapid emergence after the drug is completely shut off, and after ROC patients continue to progress to full awakening. In contrast, the scalp EEG study involved a slow emergence, as drug levels are gradually reduced in a stepwise fashion and patients are held in the emerging state for longer periods, as the drug infusion is still ongoing at low levels when patients reach ROC (Figure 4A), maintaining patients in the emerging state for longer periods. We therefore believe that these differences in arousal level around ‘ROC’, and the fact that ROC represents a somewhat different state (i.e. a prolonged plateau of low arousal in the scalp EEG case), accounts for the differences between groups.

For the difference between 4H and 4I, we apologize that it was not sufficiently clear that Figure 4H is the single subject example, whereas Figure 4I is the group analysis including all subjects. The Figure 4H panel shows data from the same single subject used in panels 4A-F, whereas the summary across subjects is in 4I. We plot both the single subject and the group because individual subjects emerge at different times throughout this long stepped infusion, and as discussed in the text only a subset of the scalp EEG subjects exhibit clear LPs. Because of this heterogeneity in timecourses, averaging the timeseries directly obscures emergence dynamics in individual subjects. We therefore show both the variance across trials within a subject (4H) and the summary statistics across all subjects (4I). We have updated the figure title and legend to make it clear that 4h displays temporal evolution of the response in one subject, whereas 4I displays the whole group. This particular subject had a slow, prolonged emergence, as can be seen by the slow return of behavioral responses over >20 minutes (Figure 4B) so that is why the LPs continue to occur 20 minutes after the defined point where ROC begins (as the subject is still undergoing the emergence process, responding to only a subset of stimuli, at this time). To clarify that the ROC+20 timepoint is still during the emergence process in this subject, we have therefore also updated the figure axis labels from "ROC+20" to "Late ROC", to clarify that the subject is still within the ROC window during this time.

2) It should be noted that Figures 4H/4I/3D all appear to quantify the same thing (LP magnitude of some sort), but the exact quantity which the authors chose to plot differs (max/mean/median amplitude). As I pointed out in my original review, it would be good to unify units as much as possible among plots.

We have now updated Figures 3 and 4 to use the same processing steps across both datasets, displaying the mean ERP value in the main figures (mean over time, then mean over trials), and we have moved the peak ERP plots to the supplementary figure (median over trials, then max value of the absolute ERP peak). Figure 4I, 3D, now show the matched mean analysis, and the new Figure 4—figure supplement 1D, E shows the matched peak analysis across the group. Figure 4H is the single subject version (i.e. no mean over trials since the boxplot itself is displaying the variance over trials within the subject).

The results were slightly clearer when using different methods for scalp vs. intracranial EEG, because the scalp data has lower temporal SNR and higher trial number, and effects are more easily detected in the peak. By contrast, the intracranial dataset has higher tSNR but does not contain enough trials for a maximum-based estimator to be robust when large slow oscillations are present, due to insufficient trials for averaging in this high-variance condition, which causes some subjects to exhibit biased large peak values. However, the overall pattern of results is similar with both of these approaches (new Figure 3D, 4I; Figure 4—figure supplement 1D, E).